# Vaccination coverage against COVID-19 among rural population in Haryana, India: A cross-sectional study

Suraj Singh Senjam[1]*, Souvik Manna[1], Garima Goel[1], Yatan Pal Singh Balhara[2], Animesh Ray[3], Yashdeep Gupta[4], Neiwete Lomi[1], Vivek Gupta[1], Praveen Vashist[1], Jeewan Singh Titiyal[1], Nitin Kashyap[1], Rajesh Kumar[1]

1 Dr. Rajendra Prasad Centre for Ophthalmic Sciences, All India Institute of Medical Sciences, New Delhi, India, 2 Department of Psychiatry, National Drug Dependence Treatment Centre, All India Institute of Medical Sciences, New Delhi, India, 3 Department of Medicine, All India Institute of Medical Sciences, New Delhi, India, 4 Department of Endocrinology, All India Institute of Medical Sciences, New Delhi, India

* drsurajaiims@gmail.com

## Abstract

### Background

Conducting a study in rural pre-dominant areas will help to understand the penetration of the vaccination campaign during the COVID-19 health crisis. This study aimed to investigate vaccination coverage against COVID-19 among the rural adult population in India and to identify factors associated with vaccination coverage.

### Methods

A population-based cross-sectional study was conducted among the rural population in one district of north India from January to February 2023. A semi-structured questionnaire was designed on the SurveyMonkey digital platform for interviewing the participants, which consisted of questions related to socio-demographic profile, health problems, vaccination status, types of vaccine, re-infection after vaccination, and functional difficulties. The data regarding infection with COVID-19 was collected based on self-reported positive testing for SARS-CoV 2 on RT-PCR.

### Findings

A total of 3700 eligible individuals were enumerated for the survey, out of which 2954 (79.8%) were interviewed. The infection rate of past COVID-19 infection, based on self-report of testing positive, was 6.2% (95%CI: 5.3–7.1). Covishield vaccine was received by most participants (81.3%, 2380) followed by Covaxin (12.3%, 361) and Pfizer manufactured vaccine (0.03,1). The coverage for first, second, and booster doses of the vaccine was 98.2% (2902), 94.8% (2802), and 10.7% (315) respectively. The risk of reinfection at 12 months or more among participants with two doses of vaccine was 1.6% (46/2802, 95%CI: 1.2–2.1). The coverage among those with severe functional difficulties was lesser as compared to those with some or no difficulties.

**Funding:** Indian Council of Medical Research, New Delhi funded (Project Grant no. 2021-6506/F1) this study. The funder does not have any role in the design and conduct of the present study. The funders had no role in study design, data collection and analysis, decision to publish, or preparation of the manuscript.

**Competing interests:** The authors have declared that no competing interests exist.

## Interpretation

Vaccination coverage against COVID-19 in rural Haryana, India is not dependent on factors like gender or occupation but is dependent on age and education. Although the full and partial vaccination coverage is high, the booster dose coverage is poor. In addition, the presence of severe disability was significantly associated with reduced vaccination coverage.

## Introduction

The World Health Organization announced on May 5, 2023, that coronavirus disease 2019 (COVID-19) is no longer an international public health emergency [1]. While this signifies a significant public health gain, the challenge of COVID-19 is far from over as it is expected that it is on its way to turning into an endemic, which means the constant, habitual, or expected presence of the disease in a particular geographic area [2]. However, a more pragmatic and optimistic view of severe acute respiratory syndrome coronavirus-2 (SARS-CoV-2), the virus that causes COVID-19 becoming endemic is that a significant proportion of the population will have achieved immunity through vaccination and/or natural infection resulting in reduced transmission and hopefully, lesser morbidity and mortality [3].

The response to the SARS-CoV-2 outbreak and the subsequent COVID-19 pandemic included developing and marketing various COVID-19 vaccines. However, inequitable access to vaccines, particularly in lower- and middle-income countries (LMICs), still poses risks of unprecedented disruptions and the emergence of viral mutations [4]. Against the backdrop of the non-availability of effective pharmaceutical management, vaccination remains the most important and cost-effective public health intervention to reduce not only the transmission and mortality due to COVID-19 but also the prevalence of post-COVID-19 symptoms and long-term complications [5]. One of the key indicators to gauge the effectiveness of the vaccination program against COVID-19 is the vaccine coverage. Indeed, the virus is not eradicated as yet, which means it can continue to spread and evolve over time. Even if COVID-19 becomes endemic, there is still potential for a resurgence with new related strains or re-infection still possible. Further vaccination has the potential to reduce the likelihood of new variants from emerging. As pandemic fatigue and reduced caution emerge among the population, maintaining good vaccine coverage would be an intuitive solution for preventing further spread or future pandemic.

Evidence demonstrates that the approved or authorized COVID-19 vaccines are both efficacious and effective against symptomatic, laboratory-confirmed COVID-19, including severe forms of the disease and death [6]. In addition, a growing body of evidence suggests that COVID-19 vaccines also reduce asymptomatic infection and transmission [7,8]. High vaccine coverage is essential in improving herd immunity, i.e. to keep vulnerable groups who cannot get vaccinated safe and protected from COVID-19 by surrounding them with a multitude of immune persons who do not transmit the disease [9]. Therefore, COVID-19 vaccination is considered one of the most important public health strategies to address COVID-19.

In January 2021, the vaccination campaign was launched by the Indian government initially for healthcare and frontline workers (HCWs), followed by the general population in a graded manner. The two vaccines available were COVAXIN, manufactured by Bharat Biotech, Hyderabad India, and COVISHIELD, developed by Serum Institute of India, Pune, Maharashtra. This was one of the largest vaccination drives in the world intending to provide two doses of vaccine, followed by a booster (precautionary) dose to all 18 years and above population

groups at public as well as private vaccination centers. The drive prioritized healthcare workers, frontline workers, and the elderly population (60 years and above) who were vulnerable to infection, intending to immunize the entire adult population. The administration of precautionary doses to the adult population (18 years and above) through private vaccination centers was started on 10 April 2022 [10]. All those who were more than 18 years of age and had completed 9 months after the administration of a second dose, were eligible for the precautionary dose.

At the time of the rollout, the supposed digital divide was a key criticism of India's vaccination policy, with the argument that rural areas would be left out, as the information technology platform could only serve those who resided in urban areas and were educated [11]. This concern gained more traction because the COVID-19 mortality rates were higher in rural areas compared to urban areas. This was attributed to factors such as skewed distribution of healthcare infrastructure, and lack of manpower and equipment, compounded with socioeconomic disadvantage of the population [12,13]. Further, there are disparities in vaccination coverage between urban and rural areas that may lead to differential mortality. Lower vaccination rates in rural areas are concerning given higher rural COVID-19 mortality rates and recent surges in cases [13]. Limited population-based studies are available on COVID-19 vaccination coverage, especially in rural areas of the Indian subcontinent. More evidence is needed to determine what level of vaccination coverage is needed to prevent the COVID-19 crisis, especially in rural India. Therefore, the current study aimed to investigate the vaccination coverage of COVID-19 among the rural adult population in India and to identify factors associated with vaccination coverage. Additionally, vaccination among the vulnerable population (persons with various functional difficulties) who are at greater risk for developing complications and comorbidities were also studied.

## Materials and methods

A population-based cross-sectional study was conducted among the rural population in northern India from January to February 2023. One predominantly rural district was selected purposively for the survey (Jhajjar, Haryana) based on logistic and administrative feasibility. Jhajjar is a predominantly rural district in the state of Haryana having a population of nearly 1 million with a sex ratio of 862 and a literacy rate of 80.7% [14]. There are four sub-districts in Jhajjar district: Badli, Bahadurgarh, Beri, and Jhajjar, out of which Jhajjar subdistrict was also selected purposively for the study due to feasibility issues. Choosing a rural area will help us to know the penetration of the vaccination drive and acceptance.

### Study questionnaire

A semi-structured questionnaire (study tool) was developed using the SurveyMonkey© digital platform for interviewing the participants. The SurveyMonkey is a cloud-based online survey software that can be employed for developing survey questionnaires and further emailed, or sent through WhatsApp or posted on a website, and shared on social media for self-administration. It can be used for face-to-face data collection using a trained interviewer like a computer-assisted personal interview (CAPI). CAPI needs face-to-face interviews for data collection by a trained interviewer. The SurveyMonkey has also a built-in basic data analysis package. The SurveyMonkey© website is protected by Trusted Site software which actively monitors for security issues like malware, malicious links, and phishing. In both platforms, the interviewer uses a digital device such as a tablet or smartphone to administer a survey questionnaire to respondents.

The data captured was directly integrated into a digital format, minimizing transcription errors that can occur with paper-based surveys. Training of team members on the digital tool and vernacular translation while interviewing the participants was standardized to reduce bias. A technical team was set up at the base hospital to troubleshoot any technical glitches such as device malfunctions or software crashes that could disrupt the survey process and lead to data loss. To ensure data security and confidentiality, data collectors were trained to clear their cache after batch upload to the central server.

The Nominal Group Technique was used to develop the questionnaire which consisted of four different stages: silent idea generation, round robin, clarification of the ideas, and voting (ranking) [15]. The final study tool consisted of a socio-demographic profile, health problems (self-reported) before the pandemic, and vaccination status, including the type of vaccine, re-infection after vaccination, and post-COVID-19 symptoms. The data regarding infection with COVID-19 was collected based on self-reported positive testing for SARS-CoV 2 on reverse transcriptase polymerase chain reaction (RT-PCR). Diagnostic test reports of RTPCR were checked, if available; but the self-report was considered sufficient for the study.

Since the infection was so unprecedented, we collected the infection history regardless of vaccination status after the demographic information. However, during the interview of the vaccination status, we categorically asked questions regarding re-infection of COVID-19 after vaccination. The information on re-infection was only after vaccination, reinfection in the absence of vaccination was excluded.

The survey tool was developed in English language and was not translated into the local language. However, the patient information sheet (PIS), provided to each participant, was in the local language (HINDI). The study team explained and read aloud each question to eligible participants in their vernacular language (Hindi) during the interview. The survey team was well-versed in English and was trained to ask questions in the local language.

The tool was pretested on a group of non-study participants, and necessary modifications were made in the semantics and language of the tool before the inception of the study. The non-study participants were selected from a rural area of Ballabgarh, Haryana, not included in the sampling frame of the main study.

We also used The Washington Group Short Set of Functioning (WG-SS) question to assess the level of difficulties in study participants. The WG-SS consists of difficulties in seeing, hearing, walking, remembering, self-care, and communication, measured with four response categories (Washington Group on Disability Statistics).

**Sample size and sampling technique.** The estimated sample size of 3700 was arrived at based on the seroprevalence of COVID-19 in the adult population (8.5%), relative precision of 15%, 80% power at 95% confidence limits, design effect of 1.65 and non-response rate of 20% [16].

A multistage cluster random sampling was used for selecting the clusters. In this method, a list of polling booths in the selected sub-district was prepared from the Election Commission of India website, which constituted the sampling frame of primary sampling units. From this sampling frame, a list of forty clusters was further selected randomly using computer software. Each polling booth (cluster) has an adult population of 1000 (500–2000) aged 18 years and above. Within each cluster, the compact segment sampling technique was used to select the households. To execute this, each cluster was divided into equal segments of approximately 20 to 25 households, each segment having a population of 70–100 individuals aged 18 years and above. A sketch map drawn with the help of local volunteers (e.g., accredited social health activist-ASHA or key local workers) was used to divide the clusters into segments. Each segment was given a serial number and one segment was selected randomly using the number of currency notes for inclusion in the survey. In the currency note method, the last digit of the

serial number on any one selected currency note is used and the segment with this same serial number becomes the selected segment. Next, the team covered all the households in the selected segment starting from one end to the other. In case the adequate number of adults (70–100) was not present in the selected segment, the adjoining segment (closest) was also covered till the required number of adults was reached.

**Data collection.**   A team comprising of a supervisor, two field investigators, and four field assistants was involved in the survey along with ASHA workers. The team moved door-to-door with the interviewer asking questions in the local language. It is well known that rural populations have poor digital literacy and many of the vaccine beneficiaries might not have access to mobile phones, especially elderly and vulnerable populations. In addition, the vaccination certificates issued by the health authorities are electronic and can be accessed from the CoWIN app. During the piloting that covered 10 households (31 eligible individuals), many participants did not have certificates despite received the COVID-19 vaccine. Depending on the e-certificates for confirming vaccination status will lead to underestimation of the true coverage, hence self-reported COVID-19 vaccine status was recorded by the interviewer. The assessment of disability was also not based on clinical examination but employed self-reported functional limitation (WGSS) as a proxy indicator for disability. All those respondents who reported some or severe difficulty in any of the six functional domains were further segregated based on their vaccination status, to generate disability, disaggregated data. However, anthropometric measurements were performed by the field assistants who were trained for the measurement of weight and height. A portable stadiometer (SECA Model 214, Seca Gmbh Co, Hamburg, Germany) and digital weighing scale (SECA Model 807, Seca Gmbh Co, Hamburg, Germany) were used for anthropometric measurements, with an accuracy of up to one decimal point.

## Study definitions

**Vaccine coverage.**   It is the proportion of the eligible population who have received a specific vaccine according to the recommended protocol. It can be calculated for the first, second, and booster doses separately. For the present study, we relied on the responses from the interviewed person.

**Partially vaccinated.**   Individual who has received a single dose of COVID-19 vaccine of any type.

**Fully vaccinated.**   Individual who has received two doses of COVID-19 vaccine of any type.

**Booster vaccinated.**   Individual who has received three doses (booster/precautionary dose) of COVID-19 vaccine of any type.

**Re-infection rate.**   Proportion of participants self-reporting RTPCR positivity at least once within 12 months after receiving two doses of any COVID-19 vaccine (i.e., fully vaccinated).

**Ethics clearance.**   The study was approved by the institute ethics committee of a tertiary care hospital, (Ref. no. IEC-260/04.03.2022, RP36/2022) and it adhered to the tenets of the Declaration of Helsinki, 2000. The study supervisors explained to each eligible participant in their vernacular language (Hindi) in the same language about the study and provided them with a hard copy of the patient information sheet (PIS) with the contact details of the investigators. Informed consent was obtained by asking the subjects about their willingness to participate, and the same was recorded on electronic forms. The survey tool itself was not translated, and surveyors were trained to ask questions in the local language.

Informed e-consent was obtained by asking the subjects about their willingness to participate, and the same was recorded on electronic forms. The electronic consent was used to avoid

close contact between the participants and the survey team who visited door-to-door. The supervisors were also responsible for re-verification visits in a few households, to ensure quality.

**Data management and analysis.**   All data were exported from the SurveyMonkey server to STATA version 15 (StataCorp 2015, Stata Statistical Software: Release 15, College Station, TX: StataCorp LP). The server was protected with a unique user ID and password for confidentiality. Data were checked and cleaned before export and descriptive analysis was done to summarize the findings. To investigate the association between independent variables and dependent variables (vaccination status), chi-square statistics were used, and the Fisher exact test was used when values in the contingency table were less than 5. Multivariate regression analysis was used to find predictors of vaccine coverage, by using the socio-demographic factors, disability, and health status as independent variables. All statistical significance was set at a p-value $<0.05$.

# Results

## Characteristics of the sample population

A total of 3700 eligible individuals were enumerated for the survey, out of which 2954 participants (79.8%) were interviewed. The main reasons for non-response among the 746 (20.2%) respondents were refusal (224, 30.0%), preoccupation with work (202, 27.0%), lack of interest (104, 13.9%), limited time (85, 11.4%), fear of COVID-19 (76, 10.2%) and locked homes (55, 7.4%) at the time of the survey. Of the total respondents, males comprised 45.2% (1335, Table 1). Around three-fourths of the participants were aged less than 55 years. While nearly half of the respondents had studied up to senior secondary or above, approximately 14.5% (427) of them were illiterate. The majority of the respondents (59.2%, 1750) were currently not in the workforce, either unemployed, retired, or homemaker and 10.5% were cultivators by occupation, whereas few respondents worked in the public (3.4%, 100) or private (11.3%, 333) sectors. The categories of retired, unemployed, and homemaker participants were clubbed together and categorized as not working groups, as also reported in previous studies on the COVID-19 vaccine [17]. Approximately, one-third of participants were overweight or obese (Table 1).

Considering the pre-existing co-morbidities of the participants, hypertension (6.1%), joint problems (4.9%), and diabetes mellitus (3.2%) were the most common underlying self-reported health problems before suffering from COVID-19. The results also showed that a total of 6.2% (183) respondents had COVID-19 disease in the past (self-reported tested positive for SARS-CoV 2). Out of six different functional difficulties included in the survey, difficulty in walking, (at least some or more) was reported to be the most common (13.1%, 418), followed by difficulty in seeing (9.9%, 291), and hearing problems (5.2%, 154) respectively (Table 2).

**COVID-19 vaccination status.**   Out of the total participants (2954) enrolled, 2902 (98.2%) were vaccinated with at least one dose of any COVID-19 vaccine. In other words, 100 participants received only one dose, 2487 received two doses, 315 received three doses and 52 participants received no vaccine. Covishield was the type of COVID-19 vaccine that had been received by most participants (81.3%, 2380) followed by Covaxin (12.3%, 361) and Pfizer (0.03,1), however, 186 (6.3%) participants were not aware of the name of the vaccine received (Fig 1). Considering the COVID-19 virus disease reported by fully vaccinated participants, the reinfection rate at 12 months or more was found to be 1.6% (46/2802, 95% CI: 1.2–2.1) for single and 0.2% (5/2802) for multiple reinfections.

**Table 1. Characteristics of the study participants along with COVID-19 vaccination coverage.**

| Characteristics | N (%) | Fully vaccinated. | Vaccine coverage % (95% CI) |
|---|---|---|---|
| **Total** | 2954 | 2802 | 94.8 (94.1–96.6) |
| **Sex** | | | |
| Male | 1335 (45.2) | 1276 (45.5) | 95.6 (94.5–96.7) |
| Female | 1619 (54.8) | 1526 (54.5) | 94.3 (93.1–95.4) |
| **Age (in years)** | | | |
| 18–25 | 589 (19.9) | 551 (19.7%) | 93.5 (91.6–95.6) |
| 26–35 | 636 (21.5) | 605 (21.6) | 95.1 (93.4–96.8) |
| 36–45 | 573 (19.4) | 544 (19.4) | 94.9 (93.1–96.7) |
| 46–55 | 450 (15.2) | 431 (15.4) | 95.8 (93.9–97.6) |
| 56–65 | 360 (12.2) | 345 (12.3) | 95.8 (93.8–97.9) |
| 66–75 | 226 (7.6) | 215 (7.7) | 95.1 (92.3–98.0) |
| Age ≥76 | 120 (4.1) | 111 (4.0) | 92.5 (87.7–97.3) |
| **Education level** | | | |
| Illiterate | 427 (14.5) | 402 (14.3) | 94.1 (91.9–96.4) |
| Primary School Certificate | 284 (9.6) | 268 (9.6) | 94.4 (91.7–97.1) |
| Middle School Certificate | 359 (12.2) | 336 (12.0) | 93.6 (91.0–96.1) |
| High School Certificate | 491 (16.6) | 459 (16.4) | 93.5 (91.3–95.7) |
| Senior Secondary School | 787 (26.6) | 751 (26.8) | 95.4 (94.0–96.9) |
| Graduate and above | 606 (20.5) | 586 (21.0) | 96.7 (95.3–98.1) |
| **Occupation** | | | |
| Not working/Out of Workforce* | 1750 (59.2) | 1646 (58.7) | 94.1 (92.9–95.2) |
| Government job | 100 (3.4) | 98 (3.5) | 98 (95.2–100.8) |
| Private job | 333 (11.3) | 322 (11.5) | 96.7 (94.8–98.6) |
| Business/Shop | 170 (5.7) | 164 (5.9) | 96.5 (93.7–99.3) |
| Agriculture | 310 (10.5) | 299 (10.7) | 96.5 (94.4–98.5) |
| Student | 270 (9.1) | 253 (9.0) | 93.7 (90.8–96.7) |
| Others | 21 (0.7) | 20 (0.7) | 95.2 (85.3–105.2) |
| **Body Mass Index** | | | |
| Underweight | 254 (8.6) | 232 (8.3) | 91.3 (87.9–94.8) |
| Normal weight | 1762 (60.0) | 1680 (60.0) | 95.3 (94.4–96.3) |
| Overweight | 742 (25.3) | 705(25.2) | 95.0 (93.4–96.6) |
| Obesity | 177(6.0) | 168 (6.0) | 94.9 (91.6–98.2) |

*This category includes the unemployed, retired, homemakers, and others not in the workforce.

Given dose-wise, the coverage for the first dose of the vaccine was 98.2% (2902, 95%CI: 97.7–98.7), for the second dose was 94.8% (2802, 95% CI: 94.0–95.6), and for the booster dose was 10.7% (315, 95%CI: 9.6–11.8, Fig 2). Based on these figures, the proportion of the vaccinated population that was partially vaccinated was 3.4% (100), that fully vaccinated was 96.6% (2802/2902) and 10.8% (315) had received booster or precautionary doses for COVID-19 also (Table 1). Among the vaccinated population, 45.5% (1276) were male respondents and 54.5% (1526, Table 1) were females. Although the coverage was higher in males (95.6%) as compared to females (94.3%), the difference was not statistically significant (p value = .104, Table 3).

The vaccination coverage increased with the age of the participants, ranging from 93.5% in the 18–25 years age group to 95.1% in those aged 66–75 years. The maximum coverage was observed among the age group 46 to 65 years (95.8%, Table 1), whereas the minimum coverage

**Table 2. Health characteristics of the participants with COVID-19 vaccination coverage.**

| Characteristics | Total sampled (2954) n (%) | Fully vaccinated (2802) n (%) | Vaccine coverage % (95% CI) |
|---|---|---|---|
| **Presence of co-morbidities (self-reported)** * | | | |
| Hypertension | 181 (6.1) | 173 (6.2) | 95.6 (92.6–98.6) |
| Joint or rheumatological diseases | 144 (4.9) | 138 (4.9) | 95.8 (92.5–99.1) |
| Diabetes Mellitus | 96 (3.2) | 91 (3.2) | 94.8 (90.3–99.3) |
| Heart diseases | 33 (1.1) | 27 (1.0) | 81.8 (67.9–95.7) |
| Asthma/Lung diseases | 33 (1.1) | 30 (1.1) | 90.9 (80.6–101.3) |
| Liver diseases | 11 (0.4) | 11 (0.4) | 100 |
| History of thyroid problems | 12 (0.4) | 12 (0.4) | 100 |
| Neurological diseases | 8(0.3) | 8 (0.3) | 100 |
| Kidney diseases | 7 (0.2) | 6 (0.2) | 85.7 (50.8–120.7) |
| Cancer | 4 (0.1) | 3 (0.1) | 75.0 (45.6–154.6) |
| Others | 49 (1.7) | 42 (1.5) | 85.7 |
| No health concerns | 2546 (86.2) | 2422 (86.4) | 95.1 (94.3–96.0) |
| **COVID-19 disease (self-reported tested positive)** | | | |
| Yes | 183 (6.2) | 171 (6.1) | 93.4 (89.8–97.1) |
| No | 2771 (93.8) | 2631 (93.9) | 94.9 (94.1–95.8) |
| **Seeing difficulty** | | | |
| No difficulty | 2663 (90.1) | 2529 (90.2) | 95.0 (94.1–95.8) |
| Some difficulty | 268 (9.1) | 251 (9.0) | 93.7 (90.7–96.6) |
| Severe or more | 23(0.8) | 22(0.8) | 95.7(86.6–104.7) |
| **Hearing difficulty** | | | |
| No difficulty | 2800 (94.8) | 2659 (94.9) | 95.0(94.2–95.8) |
| Some difficulty | 140 (4.7) | 130 (4.6) | 92.9(88.5–97.2) |
| Severe or more | 14(0.5) | 13(0.5) | 92.9(77.4–108.2) |
| **Walking/Climbing Steps difficulty** | | | |
| No difficulty | 2536 (85.9) | 2413 (86.1) | 95.1 (94.3–96.0) |
| Some difficulty | 341(11.5) | 323 (11.5) | 94.7 (92.3–97.1) |
| Severe or more | 77(2.6) | 66(2.4) | 85.7 (77.7–93.7) |
| **Cognition (Remembering difficulty)** | | | |
| No difficulty | 2814(95.3) | 2679(95.6) | 95.2 (94.4–96.0) |
| Some difficulty | 113(3.8) | 99(3.5) | 87.6 (81.4–93.8) |
| Severe or more | 27(0.9) | 24(0.9) | 88.9 (76.2–101.6) |
| **Hygiene and Self-care difficulty** | | | |
| No difficulty | 2832(95.9) | 2696 (96.2) | 95.2 (94.4–96.0) |
| Some difficulty | 94(3.2) | 84(3.0) | 89.4 (83.0–95.7) |
| Severe or more | 28(0.9) | 22(0.8) | 78.6 (62.4–94.8) |
| **Communication difficulty** | | | |
| No difficulty | 2873(97.3) | 2733(97.5) | 95.1 (94.3–95.9) |
| Some difficulty | 66(2.2) | 57(2.0) | 86.4 (77.9–94.9) |
| Severe or more | 15(0.5) | 12(0.4) | 80.0 (57.1–102.9) |

*Multiple responses are taken into consideration, so the % in the round bracket indicates a relative frequency

was found to be 92.5% among those aged 76 years and above (Table 1). Full vaccination coverage was also higher among participants who were graduates and above (96.7%, Table 1), compared to those educated up to middle school (93.6%, Table 1).

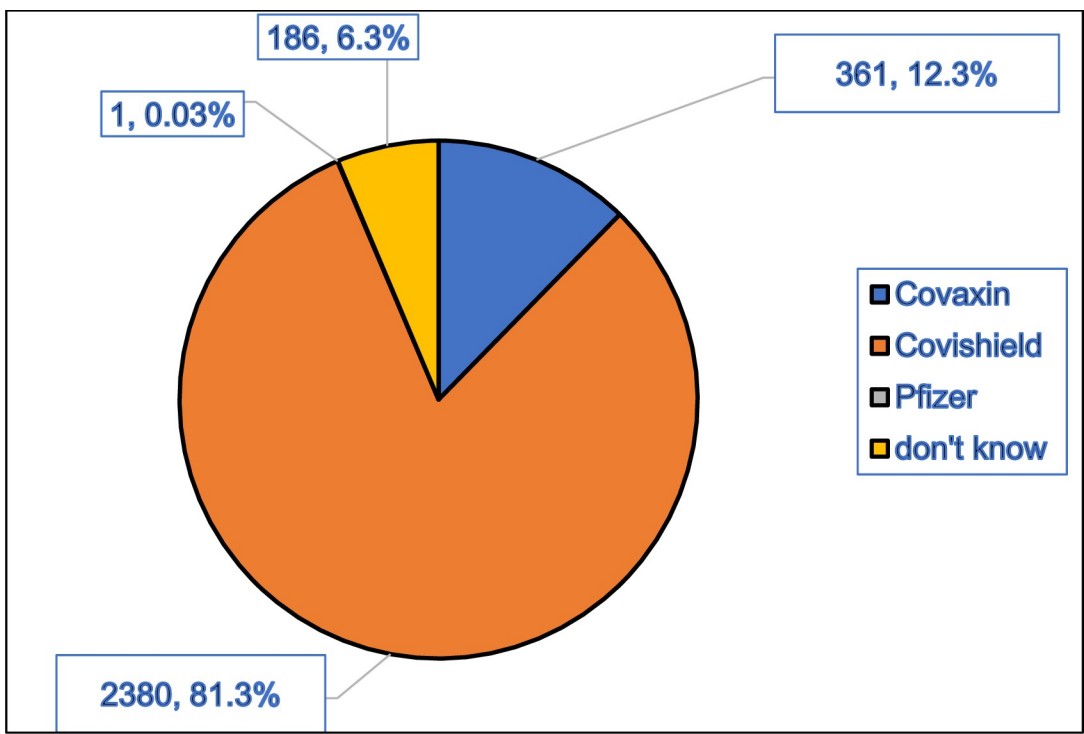

**Fig 1. COVID-19 vaccine coverage by types.**

Persons with functional difficulties in vision and hearing had better vaccination coverage, as compared to those with mobility, cognition, self-care, and communication difficulties (Table 2). The coverage in participants with severe vision problems was 95.7% (95%CI 86.6–

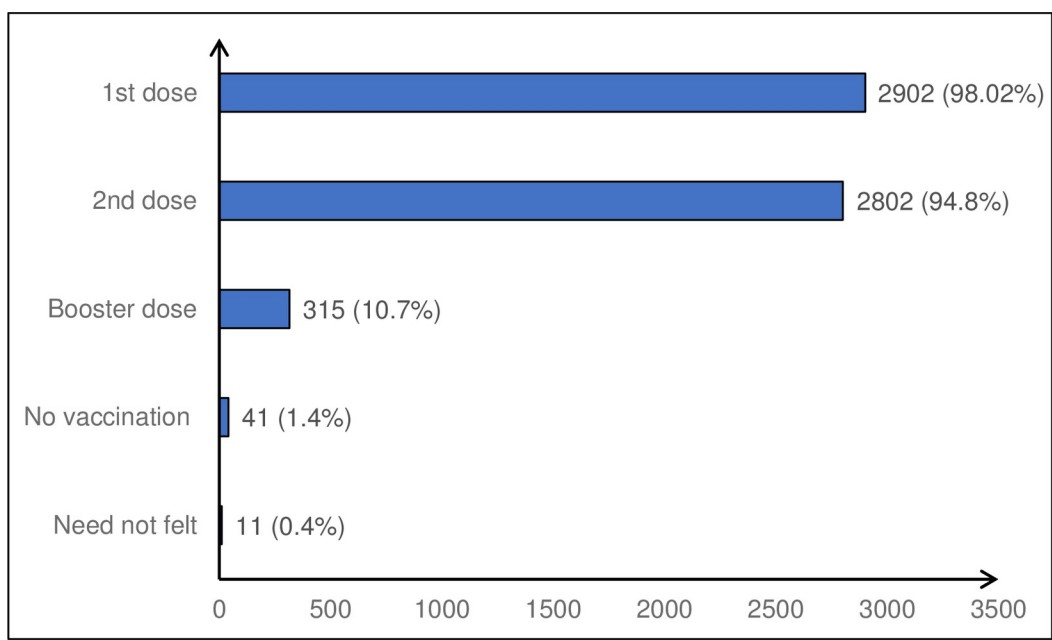

**Fig 2. COVID-19 vaccine coverage by dose wise.**

**Table 3. Factors associated with COVID-19 vaccination (N = 2954).**

| Characteristics | COVID-19 vaccination | | p-value |
|---|---|---|---|
| **Sex** | **Fully vaccinated (n = 2802) (%)** | **Not fully vaccinated (n = 152) (%)** | |
| Male | 1276 (95.6) | 59 (4.4) | 0.10 |
| Female | 1526 (94.3) | 93 (5.7) | |
| **Self-reported COVID-19 disease** | | | |
| Yes | 171 (93.4) | 12 (6.6) | 0.37 |
| No | 2631 (95.0) | 140 (5.1) | |
| **The severity of COVID-19 illness** | | | |
| Asymptomatic | 138 (94.5) | 8 (5.5) | 0.26 |
| Symptomatic | 33 (89.2) | 4 (10.8) | |
| **Post COVID symptoms** | | | |
| No | 133 (94.3) | 8 (5.7) | 0.33 |
| Yes | 33 (91.7) | 3 (8.3) | |
| Don't know | 5 (83.3) | 1 (16.7) | |
| **Seeing difficulty** | | | |
| No difficulty | 2529 (95.0) | 134 (5.0) | 0.64 |
| Some difficulty | 251 (93.7) | 17 (6.3) | |
| Severe/More | 22(95.7) | 1(4.3) | |
| **Hearing difficulty** | | | |
| No difficulty | 2659 (95.0) | 141 (5.0) | 0.51 |
| Some difficulty | 130 (92.9) | 10 (7.1) | |
| Severe/More | 13 (92.9) | 1 (7.1) | |
| **Walking/Climbing steps difficulty** | | | |
| No difficulty | 2413 (95.1) | 123 (4.9) | 0.0011 |
| Some difficulty | 323 (94.7) | 18 (5.3) | |
| Severe/More | 66 (85.7) | 11 (14.3) | |
| **Cognition (Remembering difficulty)** | | | |
| No difficulty | 2679 (95.2) | 135 (4.8) | 0.0006 |
| Some difficulty | 99 (87.6) | 14 (12.4) | |
| Severe/More | 24 (88.9) | 3 (11.1) | |
| **Hygiene** | | | |
| No difficulty | 2696 (95.2) | 136 (4.8) | 0.0003 |
| Some difficulty | 84 (89.4) | 10 (10.6) | |
| Severe/More | 22 (78.6) | 6 (21.4) | |
| **Communication** | | | |
| No difficulty | 2733 (95.1) | 140 (4.9) | 0.0010 |
| Some difficulty | 57 (86.4) | 9 (13.6) | |
| Severe/More | 12 (80.0) | 3 (20.0) | |

*Row percentages

104.7) while it was only 78.6% (95% CI 62.4–94.8) among those with difficulties in self-care. The vaccination coverage decreased as the level of functional difficulty increased, irrespective of the functional domain (Table 2). Using univariate analysis, the difference in the proportion of individuals who got vaccinated was statistically significant in the four functional domains of mobility, cognition, self-care, and communication (p value <0.001, Table 3). However, multivariate analysis did not demonstrate these significant differences in vaccination rates based on functional limitations (Table 4).

**Table 4. Predictors of COVID-19 full vaccination among study participants (N = 2954).**

| Characteristics | Fully vaccinated n (%) | Not fully vaccinated n (%) | Adjusted OR (95% CI) | p-value |
|---|---|---|---|---|
| **Sex** | | | | |
| Male | 1276 (95.6) | 59 (4.4) | - | - |
| Female | 1526 (94.3) | 93 (5.7) | 1.00 (0.64–1.55) | 0.996 |
| **Age(in years)** | | | | |
| 18–25 | 551 (93.5) | 38 (6.5) | - | - |
| 26–35 | 605 (95.1) | 31 (4.9) | 1.35 (0.76–2.39) | 0.310 |
| 36–45 | 544 (94.9) | 29 (5.0) | 1.53 (0.83–2.80) | 0.171 |
| 46–55 | 431 (95.8) | 19 (4.2) | 2.35 (1.18–4.70) | 0.016 |
| 56–65 | 345 (95.8) | 15 (4.2) | 2.95 (1.34–6.47) | 0.007 |
| 66–75 | 215 (95.1) | 11 (4.9) | 2.81 (1.14–6.94) | 0.025 |
| ≥76 | 111 (92.5) | 9 (7.5) | 2.66 (0.93–7.61) | 0.068 |
| **Education** | | | | |
| Illiterate | 402 (94.1) | 25 (5.9) | - | - |
| Primary School Certificate | 268 (94.4) | 16 (5.6) | 1.04 (0.51–2.10) | 0.913 |
| Middle School Certificate | 336 (93.6) | 23 (6.4) | 0.89 (0.45–1.73) | 0.727 |
| High School Certificate | 459 (93.5) | 32 (6.5) | 0.93 (0.48–1.80) | 0.821 |
| Senior Secondary School | 751 (95.4) | 36 (4.6) | 1.58 (0.78–3.22) | 0.204 |
| Graduate and above | 586 (96.7) | 20 (3.3) | 2.31 (1.05–5.08) | 0.037 |
| **Occupation** | | | | |
| Not working/Out of the workforce | 1646 (94.1) | 104 (6.0) | - | - |
| Government Job | 98 (98.0) | 2 (2.0) | 2.06 (0.47–8.99) | 0.338 |
| Private Job | 322 (96.7) | 11 (3.3) | 1.47 (0.71–3.04) | 0.298 |
| Business/Shop | 164 (96.5) | 6 (3.5) | 1.60 (0.63–4.04) | 0.322 |
| Agriculture | 299 (96.5) | 11 (3.5) | 1.53 (0.75–3.11) | 0.246 |
| Student | 253 (93.7) | 17 (6.3) | 0.96 (0.48–1.90) | 0.902 |
| Others | 20 (95.2) | 1 (4.8) | 1.3 (0.17–10.18) | 0.803 |
| **Self-reported COVID-19 disease** | | | | |
| Yes | 171 (93.4) | 12 (6.6) | 0.60 (0.32–1.14) | 0.120 |
| No | 2631 (94.9) | 140 (5.1) | - | - |
| **Health problems** | | | | |
| Yes | 380 (93.1) | 28 (6.9) | 0.71 (0.42–1.19) | 0.190 |
| No | 2422 (95.1) | 124 (4.9) | - | - |
| **Seeing difficulty** | | | | |
| No difficulty | 2529 (95.0) | 134 (5.0) | - | - |
| Some difficulty | 251 (93.7) | 17 (6.3) | 0.94(0.50–1.73) | 0.831 |
| Severe/More | 22 (95.7) | 1 (4.3) | 1.89 (0.23–15.64) | 0.556 |
| **Hearing difficulty** | | | | |
| No difficulty | 2659 (95.0) | 141 (5.0) | - | - |
| Some difficulty | 130 (92.9) | 10 (7.1) | 1.33 (0.54–3.30) | 0.536 |
| Severe/More | 3 (92.9) | 1 (7.1) | 1.74 (0.17–17.48) | 0.639 |
| **Walking/Climbing steps difficulty** | | | | |
| No difficulty | 2413 (95.1) | 123 (4.9) | - | - |
| Some difficulty | 323 (94.7) | 18 (5.3) | 1.17 (0.58–2.37) | 0.666 |
| Severe/More | 66 (85.7) | 11 (14.3) | 0.73 (0.23–2.35) | 0.603 |
| **Cognition (Remembering difficulty)** | | | | |
| No difficulty | 2679 (95.2) | 135 (4.8) | - | - |

*(Continued)*

**Table 4.** (Continued)

| Characteristics | Fully vaccinated n (%) | Not fully vaccinated n (%) | Adjusted OR (95% CI) | p-value |
|---|---|---|---|---|
| Some difficulty | 99 (87.6) | 14 (12.4) | 0.48 (0.19–1.26) | 0.137 |
| Severe/More | 24 (88.9) | 3 (11.1) | 1.64 (0.22–12.03) | 0.625 |
| **Hygiene** | | | | |
| No difficulty | 2696 (95.2) | 136 (4.8) | - | - |
| Some difficulty | 84 (89.4) | 10 (10.6) | 0.77 (0.27–2.19) | 0.629 |
| Severe/More | 22 (78.6) | 6 (21.4) | 0.47 (0.09–2.26) | 0.347 |
| **Communication** | | | | |
| No difficulty | 2733 (95.1) | 140 (4.90) | - | |
| Some difficulty | 57 (86.4) | 9 (13.6) | 0.48 (0.16–1.47) | 0.198 |
| Severe/More | 12 (80.0) | 3 (20.0) | 0.33 (0.04–2.47) | 0.281 |

**Predictors of COVID-19 vaccination coverage.** An adjusted analysis using multivariate regression reported that gender is not a significant predictor of COVID-19 vaccine coverage, with the aOR:1.00 (95%CI:0.64–1.55, p = 0.996). Multiple logistic regression also showed that the odds of having vaccination were significantly higher among older age groups (p < 0.005; odds ratio [OR] = 2.4; 3.0; 2.8 for age groups 46–55; 56–65; 66–75 respectively). The analysis found that participants who graduated and above showed a higher vaccination coverage rate than illiterates. Graduates and above were 2.31 times more likely to get vaccinated as compared to illiterates (aOR: 2.31; 95%CI: 1.05–5.08, p = 0.037). (Table 4).

## Discussion

The current study was done to determine vaccination coverage among rural population of north India using a population-based design. The full and partial vaccination coverage found in the current study was 94.8% and 98.2% respectively. India's indigenously developed digital platform, CoWIN, provides a live dashboard of vaccine coverage, allows every citizen the facility of conveniently and safely booking vaccine appointments, and also generates digital vaccine certificates in real-time. As per the CoWin dashboard on 12th September 2023, the full and partial coverage for India stood at 87.81% and 94.77% respectively [18]. The corresponding figures for the state of Haryana are 83.8% and 99.9% respectively [19]. Hence, the present study shows that the coverage for both single and second doses in the rural population is better than the national average, at the same time, the coverage for the second dose is better than the state average. Further study is required to investigate why this study area has a better level of coverage than the state concerned and national level. Findings from such studies may help to address other vaccination programs. However, the large vaccination coverage could be due to several factors, such as the adequate healthcare infrastructure in Jhajjar, Haryana that includes a national dedicated COVID-19 care facility assisting in vaccination in the region, active promotion of COVID-19 care services along with free vaccination programs across the state, easy access to user-friendly CoWIN vaccination app, active mass awareness campaign employing various platforms, good coordination among of all health care facilities, good connectivity leading to better transportation. Moreover, Haryana has one of the highest per capita incomes in the country.

National Institution to Transform India (NITI Ayog) has also reported that more doses have been administered in rural areas than in urban India [11]. However, it needs to be emphasized that the eligible population for vaccination in the national and state data is 12 + years, whereas the eligible population is 18+ years in the current study. There are certain

challenges in vaccinating adolescents aged 12–18 years and in determining the vaccine coverage. The most important one is that not all vaccines are authorized for administration to children and adolescents of given vaccine security. The Drug Controller General of India (DCGI) has recommended only Covaxin for children 15 to 18 years, Corbevax for children 12–14 years, and has granted emergency use authorization of Corbevax for children aged 5 to 12 years. Hence, vaccination coverage among children and adolescents is heavily dependent on the type of vaccines available and the logistics to reach every beneficiary. Another challenge in determining coverage is that the methodology for adult beneficiaries cannot be customized for younger ages as the co-morbidity, disability, and anthropometric assessments were targeted toward the adult population. As far as coverage for the booster dose (precautionary dose) is concerned, it was 20.9% for India and only 9.9% for Haryana [19]. The coverage for booster dose found in the current study was 10.7%, which is slightly better than the state average while being much lower than the national average.

## Types of vaccines

The proportion of vaccines being received found in the current study was Covaxin (12.3%), Covishield (81.3%) and Pfizer (0.03%). The proportion of vaccines administered free of cost by the government are Covishield (79.3%), Covaxin (16.5%), Sputnik V (0.06%), Corbevax (3.3%), and Covovax (0.002%) [19]. Pfizer was available only in the private sector in India and in selected states only due to their higher cost and regulatory restrictions.

**Re-infection rates after vaccination.** One of the important findings in the present study is that the reinfection rate (self-reported but tested positive) among fully vaccinated participants (who received two doses) is 1.6% at 12 months or more. The duration of 12 months between the second vaccine dose and re-infection was stipulated to avoid any overlap between post-COVID-19 syndrome (especially long COVID) and re-infection. Previous studies have demonstrated that long COVID usually starts after 3 months of infection, and the sequelae can persist even after 6 months [20]. A study from Italy reported a cumulative overall incidence of reinfection as 3.08% of those at risk for reinfection and two doses of COVID-19 vaccine reduced the risk of infection by 98% in the pre-Omicron era [21]. Another study among healthcare workers (HCWs) in India reported a reinfection rate of 7.26% (95% CI: 6.09–8.66), ranging from 18.05% (95% CI: 14.02–23.25) among unvaccinated, 15.62% (95% CI: 11.42–21.38) among partially vaccinated and 2.18% (95% CI: 1.35–3.51) among fully vaccinated HCWs [22]. The reinfection rate reported in the current study is lower than that among HCWs because it is a general population study.

Gender was not a significant predictor of vaccine coverage in the current study; previous studies had reported differences in coverage based on gender, but conclusive evidence to determine gender effect is lacking [23,24]. Older age and higher education were found to be significant predictors in this rural study area, which is also corroborated by previous studies [23]. Previous studies have reported that an increase in the proportion of people living in multidimensional poverty reduces COVID-19 vaccination coverage [25]. Also, vaccine hesitancy tends to be influenced by numerous factors like gender, education, occupation, and socio-economic status, which usually leads to a rural disadvantage [26–28].

Prevalence of self-reported COVID-19 positivity and pre-existing co-morbidities. The prevalence of COVID-19 has been reported by previous studies among healthcare workers as 11.0%, with diabetes and hypertension being the most common comorbidities [29]. Seroprevalence is measured by the presence of antibodies that signal that a person was either infected or vaccinated at some past date and shows that the body has subsequently produced detectable antibodies [30]. The national COVID-19 serosurvey was a large community-based study

conducted by the Indian Council of Medical Research (ICMR) among the general population, which reported population-weighted seroprevalence of 0.73% [95% CI: 0.34–1.13] in May 2020, and further increased to 67.6% (95% CI: 66.4–68.7) by July 2021 [31,32]. This indicates seroconversion owing to the effect of natural infection as well as vaccination [30]. The current study found a self-reported tested-positive, COVID-19 infection rate of 6.2% among the rural population, with hypertension (6.1%), joint problems (4.9), and diabetes mellitus (3.2) being the most common self-reported comorbidities.

In a previous study from two South Indian states, the infection probabilities ranged from 4.7% to 10.7% for low-risk and high-risk contact (close social contact) respectively [33]. Another study from America among 1,00,000 college students had reported a self-reported COVID-19 rate of 6.8% [34]. The self-reported SARS-CoV-2 positive rate in the current study was 6.2 (183/2954), which falls within the infection range reported from South India, as well as America.

Persons with disabilities (PwDs) are a diverse group, and the risks, barriers, and impacts faced by them will vary in different contexts according to, among other factors, their age, gender identity, types of disability, ethnicity, sexual orientation, and migration status [35]. Numerous studies have highlighted the adverse impact of the COVID-19 pandemic on the PwDs, as far as healthcare access and utilization of services are concerned [36]. A review of vaccination coverage suggested a likelihood of missed immunizations in PwDs, leading to lower rates of immunization uptake across a range of different vaccines than their non-disabled peers [37]. The current study also found a statistically significant difference in vaccination coverage among persons with mobility, cognition, self-care, and communication difficulties, as compared to their peers with no difficulty. To mitigate this inequity in coverage, further studies are needed such as qualitative study or semi-structured interview to identify the barriers that PwDs face in accessing vaccination and determine the appropriate strategies to address poor coverage."

If required, individuals in their support network such as family members, caregivers, assistants, or non-government organizations should be identified who facilitate the process to ensure that they reach and navigate the vaccination sites [35]. Environmental accessibility is one of the strategies to improve healthcare access to persons with mobility difficulties [38]. A case study from Chile shows how the primary care system can be used to provide in-home-based vaccinations for those with mobility impairments, and clinics can offer specific days for PwDs to ensure that the clinic environments can accommodate any impairments [39]. India also successfully used this strategy of home-delivery of vaccines to elderly, bed-ridden persons as well as PwDs to increase coverage in these vulnerable groups. However, equitable access remains a challenge in many underserved areas and underprivileged populations in developing nations, including India.

There are a few limitations in this study. First, it is a cross-sectional study, restricted to a single geographic location, which limits the generalizability of the study. In addition, temporal associations between vaccination and infections cannot be deduced owing to the cross-sectional nature of the study, which might require a case-control study design in the future. Second, we excluded participants aged below 18 years. Hence, the true vaccination rate among the eligible population is not reflected. Third, social desirability and recall biases are inherent due to the self-reported nature of the study. The strength of the study lies in its novelty, adequate sample size, community-based design, along disability-disaggregation of the coverage data.

## Conclusions

The COVID-19 vaccine is viewed as the most important public health measure against the SARS-CoV 2. The study concludes that coverage in rural areas of India is good, and factors

like age, gender, education, and occupation do not have a significant impact. The self-reported prevalence of COVID-19 disease is 6.2%. The full and partial vaccine coverage among the study population is 94.8% and 98.2% respectively. In addition, disability in the domains of mobility, cognition, self-care, and communication reduces vaccination coverage. The study adds evidence to the literature on vaccine coverage among rural areas of India, along with the determinants and predictors leading to good coverage.

## Supporting information

**S1 Data.**
(XLSX)

## Acknowledgments

We would like to thank all district health authorities and Accredited Social Health Activities community workers for their unconditional support in collecting the data. We appreciate all staff of Dr. Rajendra Prasad Centre for Ophthalmic Sciences, All India Institute of Medical Sciences, New Delhi who helped us in financial management and obtaining relevant official permissions.

## Author Contributions

**Conceptualization:** Suraj Singh Senjam, Yatan Pal Singh Balhara, Animesh Ray, Yashdeep Gupta, Praveen Vashist, Jeewan Singh Titiyal.

**Data curation:** Suraj Singh Senjam, Souvik Manna, Garima Goel, Vivek Gupta, Nitin Kashyap, Rajesh Kumar.

**Formal analysis:** Souvik Manna, Garima Goel.

**Funding acquisition:** Suraj Singh Senjam.

**Investigation:** Suraj Singh Senjam, Souvik Manna, Garima Goel, Yashdeep Gupta, Neiwete Lomi, Praveen Vashist, Nitin Kashyap.

**Methodology:** Suraj Singh Senjam, Souvik Manna, Yatan Pal Singh Balhara, Animesh Ray, Yashdeep Gupta, Vivek Gupta.

**Project administration:** Suraj Singh Senjam, Jeewan Singh Titiyal.

**Resources:** Neiwete Lomi.

**Software:** Nitin Kashyap, Rajesh Kumar.

**Supervision:** Animesh Ray, Praveen Vashist.

**Visualization:** Jeewan Singh Titiyal.

**Writing – original draft:** Suraj Singh Senjam, Garima Goel.

**Writing – review & editing:** Souvik Manna, Yatan Pal Singh Balhara, Animesh Ray, Yashdeep Gupta, Neiwete Lomi, Vivek Gupta.

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
