## [Decision Letter · Decision Letter 0]

14 Aug 2023

PONE-D-23-23474Vaccination Coverage against COVID-19 among a Rural Population: A Cross-Sectional Study in a Northern Part of IndiaPLOS ONE

Dear Dr. Senjam,

Thank you for submitting your manuscript to PLOS ONE. After careful consideration, we feel that it has merit but does not fully meet PLOS ONE’s publication criteria as it currently stands. Therefore, we invite you to submit a revised version of the manuscript that addresses the points raised during the review process. Please submit your revised manuscript by Sep 28 2023 11:59PM. If you will need more time than this to complete your revisions, please reply to this message or contact the journal office at plosone@plos.org. Please include the following items when submitting your revised manuscript:A rebuttal letter that responds to each point raised by the academic editor and reviewer(s). You should upload this letter as a separate file labeled 'Response to Reviewers'.A marked-up copy of your manuscript that highlights changes made to the original version. You should upload this as a separate file labeled 'Revised Manuscript with Track Changes'.An unmarked version of your revised paper without tracked changes. You should upload this as a separate file labeled 'Manuscript'.

We look forward to receiving your revised manuscript.

Kind regards,

Mosharop Hossian, MPH

Academic Editor

PLOS ONE

Journal Requirements:

Reviewers' comments:

Reviewer's Responses to Questions

**Comments to the Author**

1. Is the manuscript technically sound, and do the data support the conclusions?

Reviewer #1: Partly

Reviewer #2: Partly

Reviewer #3: Partly

Reviewer #4: Yes

Reviewer #5: Partly

2. Has the statistical analysis been performed appropriately and rigorously? 

Reviewer #1: No

Reviewer #2: No

Reviewer #3: No

Reviewer #4: No

Reviewer #5: No

3. Have the authors made all data underlying the findings in their manuscript fully available?

Reviewer #1: No

Reviewer #2: No

Reviewer #3: Yes

Reviewer #4: Yes

Reviewer #5: No

4. Is the manuscript presented in an intelligible fashion and written in standard English?

Reviewer #1: Yes

Reviewer #2: Yes

Reviewer #3: No

Reviewer #4: No

Reviewer #5: Yes

5. Review Comments to the Author

Reviewer #1: Authors have tried to see the vaccination coverage against COVID North India. Manuscript is written well. However, there are a few points to look into.

Authors should explain how this COVID vaccination coverage study is relevant to the present context , when people already accustomed with COVID , it may be a self limiting disease present in endemic form.

In methodology sampling frame , sample size has been explained very well. However authors can explain more about the sampling frame like which parts of North India. What is the basis of choosing these regions.

Lot Quality assurance sampling methodology and data analysis there of could also have been tried

In relation to data analysis, whether authors can try univariate and multivariate regression analysis rather than just a chisquare test.

Iny opinion the manuscript needs a major over haul.

Reviewer #2: The study topic is an interesting one but there are some major loopholes is there, so please look into the comments. Also, Discussion section has not been written in proper order, so please look into the section to make the paper a systematic one.

The comments are below:

1. Short title should be modified, not appropriate as its current version.

2. What is the purposive reason for selecting the district? Mention all the reasons in detail for selection.

3. Non-response rate is too high for the study? What are the major reasons for non-responses (more than 20%?)

4. If the survey has taken place offline, why their vaccine certificate has not been examined rather than relying on the responses from the interviewed person?

5. Does the study measure the BMI or is it self-reported? Mention in detail. (Table 1)

6. Does there are any clinical evidence of hearing, vision, walking, Hygiene, communication, or cognition problem? If not kindly mention it. Also, explain if it is not clinically diagnosed then these statistics are how reliable, or any validity test has been performed or not. (Table 2)

7. Did the author check vaccine-wise differences of comorbidity or any selected difficulties? If not, what are the possible justifications?

8. Minor modification is required in the first sentence of the Discussion section.

9. What is the reason for the exclusion of 12–18-year people, which share a big proportion in the population share?

Reviewer #3: The research was carried only in one sub-district of Haryana state but the title is depicting North India. The basis of survey from one district cannot depict the whole North Indian population. North India means Uttar Pradesh, Delhi, Haryana, Himachal, Jammu and Kashmir, Uttrakhand, Punjab, Ladakh and they carry almost 1/4th of India’s population which will be roughly around 32 crores. Though the author had mentioned these factors in the limitation of the study still there is a need to change its title as this cannot be extrapolated for whole North India.

The author need to explain about “Who were the non-study participants” with there academic and geo-graphical background, is it is representing rural or urban population.

The currency note method need to be elaborated. Farmers cannot be clubbed with unemployed and in table, unemployed/homemaker and retired are clubbed which is different from paragraph.

Are the COVID-19 self reporting was based on any diagnostics test or on the basis of symptoms only.

The abstract need to be re-written

“The coverage in participants with severe vision problems was 95.7%, while it was only 78.6% in those with difficulties in self-care. The coverage among those with severe difficulties in mobility, cognition, self-care, and communication were significantly lower as compared to those with no such difficulties.”- The author is emphasizing this particular finding which does not look so significant that it should be highlighted in abstract still if authors feel than they can go ahead with a shorten version.

Table 2- the health statistics of participants are pre-vaccination figures which needs to be mentioned for more clarity so as it doesnot get confused with post-vaccination complications.

The table No. 3 needs proper explanation for the data in brackets and in the footnote please mention what the bracketed data represents.

“Also, vaccine hesitancy was usually higher in urban dwellers as compared to rural population”- this statement need further support. And please go through Joshi et al., 2022

Joshi A, Surapaneni KM, Kaur M, Bhatt A, Nash D, El-Mohandes A. A cross sectional study to examine factors influencing COVID-19 vaccine acceptance, hesitancy and refusal in urban and rural settings in Tamil Nadu, India. PLoS One. 2022 Jun 9;17(6):e0269299. doi: 10.1371/journal.pone.0269299. PMID: 35679313; PMCID: PMC9182563.

“Seroprevalence is measured by the presence of antibodies that signal that a person was infected at some past date and shows that the body has subsequently produced detectable antibodies”- This statement can be seen again as what about the sero-positivity due to post-vaccination antibodies.

“In a previous study from south India among 575,071 individuals exposed to 84,965 confirmed COVID-19 cases, the infection probabilities ranged from 4.7% to 10.7% for low risk and high-risk contact types, respectively. (30) The result of the present study also falls within this infection range”- _re-word this sentence

Some reference suggested for reading and as per suitability can include

Agarwal SK, Naha M. COVID-19 Vaccine Coverage in India: A District-Level Analysis. Vaccines (Basel). 2023 May 5;11(5):948. doi: 10.3390/vaccines11050948. PMID: 37243052; PMCID: PMC10221184.

Dash P, Mansingh A, et al.,. Infection, cases due to SARS-CoV-2 in rural areas during early COVID-19 vaccination: findings from serosurvey study in a rural cohort of eastern India. Epidemiol Infect. 2022 Mar 3;150:e58. doi: 10.1017/S0950268822000346. PMID: 35287778; PMCID: PMC8937583.

Mundackal R, Agarwal T, Murali K, Isaac NV, Hu P, Dhayal V, Mony PK. Prevalence & correlates of COVID-19 vaccine hesitancy in a rural community of Bengaluru district, southern India: A preliminary cross-sectional study. Indian J Med Res. 2022 May-Jun;155(5&6):485-490. doi: 10.4103/ijmr.ijmr_3593_21. PMID: 36348594; PMCID: PMC9807210.

Murhekar MV, et al., ICMR serosurveillance group. Seroprevalence of IgG antibodies against SARS-CoV-2 among the general population and healthcare workers in India, June-July 2021: A population-based cross-sectional study. PLoS Med. 2021 Dec 10;18(12):e1003877. doi: 10.1371/journal.pmed.1003877. PMID: 34890407; PMCID: PMC8726494.

Abedin M, et al. Willingness to vaccinate against COVID-19 among Bangladeshi adults: Understanding the strategies to optimize vaccination coverage. PLoS One. 2021 Apr 27;16(4):e0250495. doi: 10.1371/journal.pone.0250495. PMID: 33905442; PMCID: PMC8078802.

Reviewer #4: Page 10 Section Introduction- Can authors describe some India-specific stats instead of LMICs, as India is no more an LMIC.

Page 11 Grammar correction- This concern gained more traction because of the fact- instead of gains.

Page 11 objective- Authors say that they want to identify factors associated with vaccination coverage, but they do not plan a statistical analysis (in the methods section) for the same at the onset of the study ab initio. They have not planned regression analysis and not addressed the low count in cells of contingency tables below 5.

It does not matter whether we get any significant factors on bivariate analysis, a plan should be in place for multivariate analysis.

Page 12 Study tool- How is SurveyMonkey platform similar/dissimilar to CAPI?

Page 12 Study tool- Authors voice a concern in the introduction section, that digital platform COWIN (which was also facilitated by others and needed connectivity) could not have been used rural areas and then they use digital questionnaire on Survey Monkey. The enablers like surveyors, cache in the device, batch upload to a central server etc, need to be highlighted comparing the same with survey using printed questionnaire.

Page 12 Survey tool- was the tool itself translated in a vernacular language, was the consent part read in a vernacular language? How far was the interaction of the participants with the platform itself (particularly for consent), although it may not have been feasible in COVID-19 pandemic or post-pandemic.

Page 12 sample size and sampling technique- for an adult population above 18 years of age, 500 to 2000 count in 20 to 25 segments, each segment can have as low population as 25 or 50 and the range can be 25 to 100. Was there any contingency planned in case the sample size was not reached in 40 clusters, like enrolling more segments/clusters (at least planning done even if it was not needed eventually).

Page 12 Sample size calculation- revise the statement for clarity as- "One segment was selected randomly using currency notes for inclusion in the survey"and the team covered all the households in the selected segment".

Page 13 Study definitions- please clarify and add- "For the present study vaccine coverage was calculated for second dose".

Page 13 Ethics clearance- English correction- Please revise the statement as "Electronic consent was used to avoid close contact between the participants and the survey team who visited door-to-door".

Page 13 Data collection- How were the responses recorded in the survey monkey? What was done to ensure that there was no variation or difference in interpretation when different interviewers asked questions in their respective segments?What checks were deployed to ensure quality of data collection? Were there re-verification visits in a few households? Were there any measures taken within the Survey monkey platform?

Page 14 Data management and analysis- In case >20% of cells had an expected count of less than 5, what tests were used? Was there no plan for multivariate analysis like regerssion? Having a homogenous population from the same segment is not a valid justification for not planning regression, since the purpose here is to find out the association after cancelling the effect of each factor on another.

Page 14 Results- Characteristics of the sample population- Please put a dot/full stop after (1619, Table 1).

Page 14 Results- Characteristics of the sample population- In the line "Approximately, two quarters

of participants..." please replace the word culivators with retired.

Page 15 In the statement- "Considering the clinical characteristics..., please put % symbol after 4.9 and 3.2. This consistency needs to be checked throughout the manuscript.

Page 15 In the statement and throughout the manuscript- were the COVID-19 infections reported to have occured before vaccination or after. Similarly, for re-infections also, temporality needs to be explained- whether there was first infection before vaccination and another after or else-wise.

Page 15- Given the statements in the introduction about a higher prevalence and death in rural areas due to COVID-19, what can be the possible explanation regarding a low rate of COVID-19 infection found by the study as 6.2%?

Page 17 COVID-19 vaccination status- for clarity, add the text- 100 participants received only one dose and 52 participants received no vaccine.

Page 17 COVID-19 vaccination status- In the statement- Considering the COVID-19 virus disease..., please replace participants with fully vaccinated (grammatically wrong), with fully vaccinated participants.

Page 18 In the statement- Persons with functional difficulties in vision- Revise the table number in the brackets as Table 2.

Page 18 Table 3- Provide only significant p values in bold and other in non-bold font. Revise the column heading/cohort name of not vaccinated as - Not fully vaccinated- since unvaccinated are only 52, while 100 are partially vaccinated- hence the 100 partially vaccinated are not "not vaccinated".

In first line of Discussion- Page 19- remove the left over "t".

To put the things in the perspective, add to the discussion- the reported national coverage for rural population?

Page 19- Discussion section- In the statement- Hence, the present study shows that the coverage... revise the text as- at the same time, the coverage for the second dose is better than the state average.

Page 19, the statement "The proportion of vaccines being received found..." the statement mentions corresponding values but the same is missing for Pfizer vaccine, and can we name this vaccine the way we have done for other vaccines?

Discussion- global comment- clarity is needed regarding the post-vaccination infections and second episodes of infection. Also need to explain the value addition of the study besides finding out vaccine coverage. Also, was it highlighted that the seroprevalence studies undertaken by other agencies will be affected by vaccine coverage as well. Temporality of the infections and vaccinations need to be clarified even if these participants were fully vaccinated. Another thing is highlighting the comparison between unvaccinated, partially vaccinated, and fully vaccinated individuals.

Page 20- comparison of the study with other studies- How do we compare the results of this study with the following systematic review with pooled seroprevalence of 20 to 70 %?

Jahan N, Brahma A, Kumar MS, Bagepally BS, Ponnaiah M, Bhatnagar T, Murhekar MV. Seroprevalence of IgG antibodies against SARS-CoV-2 in India, March 2020 to August 2021: a systematic review and meta-analysis. Int J Infect Dis. 2022 Mar;116:59-67. doi: 10.1016/j.ijid.2021.12.353. Epub 2021 Dec 28. Erratum in: Int J Infect Dis. 2022 Jun;119:119. PMID: 34968773; PMCID: PMC8712428.

Page 20- Put % sign in the brackets in the statement- joint problems (4.9) and diabetes mellitus (3.2) being the most common self-reported comorbidities.

Page 21- In the statement- To mitigate this inequity in coverage...- What were the barriers found out in the current survey for vaccine coverage in PwDs? Was there any qualitative data collected in the survey and can we at least highlight the barriers found?

Page 21- a statement of not finding the significance for any factor needs to be made regarding association. Please refer to the earleir comments.

Page 22- Conclusions- in the first statement- remove "of" from the statement- viewed as the most important public health measure of against the

SARS-CoV 2.

Reviewer #5: Dear author, there are some issues that needs to included and being clarified

1. As per the topic name of the article: Vaccination Coverage against COVID-19 among a Rural Population: A Cross, Sectional Study in a Northern Part of India, the more focus should have been given to the village conditions like transportation, electricity, restrictions etc. that differentiate the village and urban infrastructure and highly affect accessibility to basic health care services and the vaccination coverage. Also cross check the title of the article as you are saying it is the cross sectional study but your retracting the data of the past in this study that is objective of case control study.

2. This study is conducted in a very small geographic area so the significance of the results cannot be generalized.

3. The base seroprevalance for sample size calculation in the study was 8.5% , there is a need to specify the source or reference of this seroprevalence of COVID-19 in the adult population, as mentioned in the subheading "Sample size and Sampling technique". In the discussion part, you have mentioned the sero prevalence being 0.73 % in India. There seems a mismatch in both the statements and needs clarification.

4. In Covid 19 Vaccine status subheading, it is mentioned that "Covishield was the type of COVID-19 vaccine that had been received by most participants (81.3%, 2380) followed by Covaxin (12.3%, 361) and Pfizer (0.03, 1)" . The source of Pfizer Vaccine should be mentioned as it was not granted clearance by Indian regulatory authorities.

5. A comparison between the vaccination coverage (full and partial) at the national and state (Haryana) level data available on CoWin dashboard on 28th June 2023 is made with the data of the study. It is better to also compare the study data with the CoWin Dashboard vaccination coverage of study area. The reasons for the difference found, if any might be speculated.

6. The vaccination coverage in study area is better than state as well as national level coverage. So, various steps taken up by authorities like mass campaigns, setups, infrastructure and protocols for COVID19 vaccination in your study area could be compared and explained in your article. this would be act as a reference for other disease vaccination programmes and vaccination programmes to be conducted in other areas too.

7. The statement on the last paragraph of page no 13, the vaccine status of current study was compared with national status and corresponding values were mentioned viz “The proportion of vaccines being received found in the current study was Covaxin (12.3%), Covishield (81.3%) and Pfizer (0.03%). The corresponding values for the national level are Covishield (79.3%), Covaxin (16.5%) and Sputnik V (0.06%) respectively. Corresponding values for sputnik v in the study as well as Pfizer vaccine at national level should be mentioned.

8. A clarification regarding why in the study, there is reporting of the reinfection of COVID 19 among only fully vaccinated people, that too only after 12 months or more, post vaccination. The data regarding reinfection before 12 months post vaccination should also be mentioned. The comparative statement of difference between reinfection after partial vaccination and full vaccination would also be useful and help in estimating the relative Risk between these two groups.

9. Reinfection rate as per type of vaccine should also be mentioned.

10. The vaccine effectiveness in terms of reduced COVID19 cases and deaths pre and post vaccination should be taken up in the study.

11. There is a difference of 3.4 % between partial vaccination and full vaccination, the data regarding various reasons for partial vaccination should be mentioned (like: as mentioned in the article that the hesitancy for covid 19 vaccination is more in urban population compared to rural population).

12. The proportion of partially vaccinated is 3.4 %, the data regarding theses partially vaccinated people should be correlated with various variables like sex, occupation, educational status and disease literacy, etc.

13. The reinfection rate of 1.6% for single infection and 0.2 % of multiple infection is reported in the study, is there any underlying factor (health problems or occupation) for the reinfections reported.

14. One of the major limitation is that it solely dependent on the individual response that is susceptible for recall biasness and social desirability bias, so you have to give an explanation of the method you adopted for removal or reduction of these biasness in your data.

15. The data regarding aefi, if any should also be included in the study.

16. The time line in which the vaccination has been achieved in rural areas and urban areas should be compared.

17. Please correct Reference no 5., 19, 20, 22, 23, 24; as per journal guidelines.

6. PLOS authors have the option to publish the peer review history of their article (what does this mean?). If published, this will include your full peer review and any attached files.

Reviewer #1: No

Reviewer #2: **Yes: **Sourav Dey

Reviewer #3: No

Reviewer #4: **Yes: **Dr Manish Gehani

Reviewer #5: **Yes: **Baleshwari Dixit

---

## [Author Response · Author response to Decision Letter 0]

31 Oct 2023

The authors would like to thank all reviewers and editors for their time spent reading the manuscript and for helping us to improve the manuscript. We have revised as per comments and suggestions made to the manuscript. The revisions are highlighted in the revised manuscript. 

Reviewer 1

1. Authors have tried to see the vaccination coverage against COVID North India. The manuscript is written well. However, there are a few points to look into. Authors should explain how this COVID vaccination coverage study is relevant to the present context, when people already accustomed with COVID, it may be a self-limiting disease present in endemic form.

Reply: 

Thank you to the reviewer for this important comment. Indeed, the virus is not eradicated as yet, which means it can continue to spread and evolve over time. Even if COVID-19 becomes endemic, there is still potential for a resurgence with new related strains or re-infection still possible. 

The vaccine can reduce the severity of the disease and death among us, including highly vulnerable groups E.g., older and people with underlying health conditions. 

Further vaccination has potential to reduce the likelihood of new variants from emerging. (WHO)

2. In the methodology sampling frame , sample size has been explained very well. However authors can explain more about the sampling frame like which parts of North India. What is the basis of choosing these regions.

Reply:

The sentence added, “One predominantly rural district was selected purposively for the survey (Jhajjar, Haryana) based on logistic and administrative feasibility. Jhajjar is a predominantly rural district in the state of Haryana having a population of nearly 1 million with a sex ratio of 862 and a literacy rate of 80.7%. There are four sub-districts in Jhajjar district: Badli, Bahadurgarh, Beri and Jhajjar, out of which Jhajjar subdistrict was also selected purposively for the study due to feasibility issues.”

Moreover, we felt that rural penetration in the vaccination drive would help us to know the actual coverage and vaccine acceptance

3. Lot Quality assurance sampling methodology and data analysis there of could also have been tried

Reply: 

Thanks to the reviewer for the suggestions. The Lot Quality Survey can also be used to measure coverage, but rather than providing a ‘point’ estimate of coverage, it provides information on whether or not a certain level of coverage has been achieved. Investigators need to predefine a minimum level of acceptable coverage and an upper level of desirable or targeted coverage against which the lot will be assessed. This target coverage for COVID-19 vaccination is not well defined, hence LQAS was not used in the current study.

4. In relation to data analysis, whether authors can try univariate and multivariate regression analysis rather than just a chisquare test.

Iny opinion the manuscript needs a major over haul.

Reply: 

Table 4 with multivariate regression added in the results section.

Reviewer 2.

1. The study topic is an interesting one but there are some major loopholes is there, so please look into the comments. Also, Discussion section has not been written in proper order, so please look into the section to make the paper a systematic one

Reply:

The discussion section has been systematized.

2. Short title should be modified, not appropriate as its current version.

Reply: 

Short title has been modified to, “COVID-19 vaccine coverage among rural adults”

3. What is the purposive reason for selecting the district? Mention all the reasons in detail for selection.

Reply: 

Methods section: Sentence added, “One predominantly rural district was selected purposively for the survey (Jhajjar, Haryana) based on logistic and administrative feasibility. Jhajjar is a predominantly rural district in the state of Haryana having a population of nearly 1 million with a sex ratio of 862 and a literacy rate of 80.7%.(Jhajjar Population 2023, n.d.) There are four sub-districts in Jhajjar district: Badli, Bahadurgarh, Beri and Jhajjar, out of which Jhajjar subdistrict was also selected purposively for the study due to feasibility issues.” 

Moreover, we felt that rural penetration in the vaccination drive would help us to know the actual coverage and vaccine acceptance.

4. Non-response rate is too high for the study? What are the major reasons for non-responses (more than 20%?)

Reply: 

The non-response rate was 20.2% in the current study, which is slightly more than the assumed 20% for sample-size calculation. Reasons added in the result section as, “The main reasons for non-response among the 746 (20.2%) respondents were refusal (224, 30.0%), preoccupation with work (202, 27.0%), lack of interest (104, 13.9%), lack of time (85, 11.4%), fear of COVID-19 (76, 10.2%) and locked homes (55, 7.4%).”

5. If the survey has taken place offline, why their vaccine certificate has not been examined rather than relying on the responses from the interviewed person?

Reply: 

We agreed with the review comments. Initially, we thought the team could check the vaccine certificate too (in piloting). Later it was not feasible in the rural area. Many of them in rural areas do not have it despite being vaccinated. Finally we decided to rely on the responses.

Sentence added in the methods section, “It is well known that rural population have poor digital literacy and many of the vaccine beneficiaries might not have access to mobile phones or computer, especially elderly and vulnerable population. In addition, the vaccination certificates issued by the health authorities are electronic, that can be accessed from the CoWIN app. Depending on such e-certificates for confirming vaccination status may lead to underestimation of the true coverage, hence self-reported COVID-19 vaccine status was recorded by the interviewer.”

6. Does the study measure the BMI or is it self-reported? Mention in detail. (Table 1)

Reply:

It was measured in the field.

The sentence added, “However, anthropometric measurements were performed by the field assistants who were trained for measurement of weight and height. A portable stadiometer and digital weighing scale were used for anthropometric measurements, with an accuracy up to one decimal point.”

7. Does there are any clinical evidence of hearing, vision, walking, Hygiene, communication, or cognition problem? If not kindly mention it. Also, explain if it is not clinically diagnosed then these statistics are how reliable, or any validity test has been performed or not. (Table 2)

Reply: 

Sentence added in methods section, “The assessment of disability was also not based on clinical examination but employed self-reported functional limitation using WGSS as a proxy indicator for disability. All those respondents who reported some or severe difficulty in any of the six functional domains were further segregated based on their vaccination status, in order to generate disability, dis-aggregated data”. WGSS is a validated tool and has been widely used for studies conducted in India as well

8. Did the author check vaccine-wise differences of comorbidity or any selected difficulties? If not, what are the possible justifications?

Reply; 

The authors aimed to find whether vaccine coverage and the proportion of vaccinated individuals were influenced by disability. 

The sentence added in the results section, “The vaccination coverage decreased as the level of functional difficulty increased, irrespective of the functional domain. (Table 2) Using univariate analysis (Table 3), the difference in the proportion of individuals who got vaccinated was statistically significant in the four functional domains of mobility, cognition, self-care, and communication (p value <0.001). However, multivariate analysis did not demonstrate these significant differences in vaccination rates based on functional limitations. (Table 4)”

9. Minor modification is required in the first sentence of the Discussion section.

Reply: 

Sentence modified, “The current study was done to determine vaccination coverage among the rural population of North India using a population-based design.”

10. What is the reason for the exclusion of 12–18-year people, which share a big proportion in the population share?

Reply: 

Sentence added, “There are certain challenges in vaccinating adolescents aged 12-18 years and in determining the vaccine coverage. The most important one is that not all vaccines are authorized for administration to children and adolescents in view of vaccine security. The Drug Controller General of India (DCGI) has recommended only Covaxin for children 15-18 years, Corbevax for children 12-14 years and has granted emergency use authorization of Corbevax for children aged 5 to 12 years. Hence, vaccination coverage among children and adolescents is heavily dependent on the type of vaccines available and the logistics to reach every beneficiary. Another challenge in determining coverage is that the methodology for adult beneficiaries cannot be customized for younger ages as the co-morbidity, disability, and anthropometric assessments were targeted towards the adult population.”

Reviewer 3.

1. The research was carried only in one sub-district of Haryana state but the title is depicting North India. The basis of survey from one district cannot depict the whole North Indian population. North India means Uttar Pradesh, Delhi, Haryana, Himachal, Jammu and Kashmir, Uttrakhand, Punjab, Ladakh and they carry almost 1/4th of India’s population which will be roughly around 32 crores. Though the author had mentioned these factors in the limitation of the study still there is a need to change its title as this cannot be extrapolated for whole North India

Reply:

I agreed with the reviewer. In fact, we would like to indicate the place of study being conducted rather than reflecting the study covered. But we have changed the title. 

The title has been changed to, “Vaccination Coverage against COVID-19 among Rural Population in Haryana, India: A Cross-Sectional Study”

 2. The author needs to explain about “Who were the non-study participants” with there academic and geo-graphical background, is it is representing rural or urban population.

Reply: 

It was conducted in rural of Ballabgarh, Haryana. The sentence added in methods, “The non-study participants were selected from a rural area of Haryana, not included in the sampling frame of the main study.”

3. The currency note method need to be elaborated. Farmers cannot be clubbed with unemployed and in table, unemployed/homemaker and retired are clubbed which is different from paragraph.

Reply:

To ensure the random selection of a segment, we employed the number of currency notes. 

Sentences added, “Each segment was given a serial number and one segment was selected randomly using currency notes for inclusion in the survey. In the currency note method, the last digit of the serial number on any one selected currency note is used and the segment with the same serial number becomes the selected area. Next, the team covered all the households in the selected segment starting from one end to the other.”

Table 1 paragraph modified, “Majority of the respondents (59.2%, 1750) were currently not in the workforce, either unemployed, retired or homemaker and 10.5% were cultivators by occupation, whereas few respondents worked in the public (3.4%, 100) or private (11.3%, 333) sectors. The categories of retired, unemployed and homemaker participants were clubbed together and categorized as not working group, as also reported in previous studies on COVID-19 vaccine”

4. Are the COVID-19 self-reporting was based on any diagnostics test or on the basis of symptoms only.

Reply: 

Sentence added, “The data regarding infection with COVID-19 was collected based on self-reported positive testing for SARS-CoV 2 on reverse transcriptase polymerase chain reaction (RT-PCR). Diagnostic test reports of RT-PCR were checked, if available; but the self-report was considered sufficient for the purpose of the study. “

5. The abstract need to be re-written. The coverage in participants with severe vision problems was 95.7%, while it was only 78.6% in those with difficulties in self-care. The coverage among those with severe difficulties in mobility, cognition, self-care, and communication were significantly lower as compared to those with no such difficulties.”- The author is emphasizing this particular finding which does not look so significant that it should be highlighted in abstract still if authors feel than they can go ahead with a shorten version.

Reply: 

We really appreciate the reviewer for a such good suggestion. 

The sentence was shortened as “The coverage among those with severe functional difficulties was lesser as compared to those with some or no difficulties.”

6. Table 2- the health statistics of participants are pre-vaccination figures which needs to be mentioned for more clarity so as it doesnot get confused with post-vaccination complications.

Reply; 

Table 2 paragraph rephrased as, “Considering the pre-existing co-morbidities of the participants, hypertension (6.1%), joint problems (4.9), and diabetes mellitus (3.2) were the most common underlying self-reported health problems before suffering from COVID-19.”

7. The table No. 3 needs proper explanation for the data in brackets and in the footnote please mention what the bracketed data represents.

“Also, vaccine hesitancy was usually higher in urban dwellers as compared to rural population”- this statement need further support. And please go through Joshi et al., 2022

Joshi A, Surapaneni KM, Kaur M, Bhatt A, Nash D, El-Mohandes A. A cross sectional study to examine factors influencing COVID-19 vaccine acceptance, hesitancy and refusal in urban and rural settings in Tamil Nadu, India. PLoS One. 2022 Jun 9;17(6):e0269299. doi: 10.1371/journal.pone.0269299. PMID: 35679313; PMCID: PMC9182563.

Reply:

Footnote added. It is a row percentage. 

Discussion sentence modified, “Also, vaccine hesitancy tends to be influenced by numerous factors like gender, education, occupation, and socio-economic status, which usually leads to a rural disadvantage.”

8. Seroprevalence is measured by the presence of antibodies that signal that a person was infected at some past date and shows that the body has subsequently produced detectable antibodies”- This statement can be seen again as what about the sero-positivity due to post-vaccination antibodies.

Reply:

Sentence modified, “Seroprevalence is measured by the presence of antibodies that signal that a person was either infected or vaccinated at some point of time in the past date and shows that the body has subsequently produced detectable antibodies.”

9. In a previous study from south India among 575,071 individuals exposed to 84,965 confirmed COVID-19 cases, the infection probabilities ranged from 4.7% to 10.7% for low risk and high-risk contact types, respectively. (30) The result of the present study also falls within this infection range”- _re-word this sentence

Reply;

Sentence modified, The infection probabilities for low-risk to high-risk contact varied from 4.7% to 10.7% in the previous studies from two South Indian states. The self-reported SARS-CoV-2 positive rate in the current study was 6.2% (183/2954), which falls within the infection range reported from South India.”

10. Some reference suggested for reading and as per suitability can include

Agarwal SK, Naha M. COVID-19 Vaccine Coverage in India: A District-Level Analysis. Vaccines (Basel). 2023 May 5;11(5):948. doi: 10.3390/vaccines11050948. PMID: 37243052; PMCID: PMC10221184.

Dash P, Mansingh A, et al.,. Infection, cases due to SARS-CoV-2 in rural areas during early COVID-19 vaccination: findings from serosurvey study in a rural cohort of eastern India. Epidemiol Infect. 2022 Mar 3;150:e58. doi: 10.1017/S0950268822000346. PMID: 35287778; PMCID: PMC8937583.

Mundackal R, Agarwal T, Murali K, Isaac NV, Hu P, Dhayal V, Mony PK. Prevalence & correlates of COVID-19 vaccine hesitancy in a rural community of Bengaluru district, southern India: A preliminary cross-sectional study. Indian J Med Res. 2022 May-Jun;155(5&6):485-490. doi: 10.4103/ijmr.ijmr_3593_21. PMID: 36348594; PMCID: PMC9807210.

Murhekar MV, et al., ICMR serosurveillance group. Seroprevalence of IgG antibodies against SARS-CoV-2 among the general population and healthcare workers in India, June-July 2021: A population-based cross-sectional study. PLoS Med. 2021 Dec 10;18(12):e1003877. doi: 10.1371/journal.pmed.1003877. PMID: 34890407; PMCID: PMC8726494.

Abedin M, et al. Willingness to vaccinate against COVID-19 among Bangladeshi adults: Understanding the strategies to optimize vaccination coverage. PLoS One. 2021 Apr 27;16(4):e0250495. doi: 10.1371/journal.pone.0250495. PMID: 33905442; PMCID: PMC8078802.

Reply: 

Suggested references are added in appropriate sections.

Reviewer 4.

1. Page 10 Section Introduction- Can authors describe some India-specific stats instead of LMICs, as India is no more an LMIC.

Reply: As per World Bank, India is still a lower and middle-income country, and has not yet been designated as an upper-income country.

2. Page 11 Grammar correction- This concern gained more traction because of the fact- instead of gains.

Reply: Correction done.

3. Page 11 objective- Authors say that they want to identify factors associated with vaccination coverage, but they do not plan a statistical analysis (in the methods section) for the same at the onset of the study ab initio. They have not planned regression analysis and not addressed the low count in cells of contingency tables below 5.

Reply: 

Thank you to the reviewer. we have added this in the data analysis as, “To investigate the association between independent variables and dependent variables (vaccination status) chi-square statistics were used, and Fisher exact test was used when values in contingency table were less than 5. Multivariate regression analysis was used to find predictors of vaccine coverage, by using the socio-demographic factors, disabilities and health status as independent variables.”

Table 4 has been added.

4. It does not matter whether we get any significant factors on bivariate analysis, a plan should be in place for multivariate analysis.

Reply: Yes, very true what the reviewer suggested. Thank you so much. Table 4 with multivariate regression added in the results section.

5. Page 12 Study tool- How is SurveyMonkey platform similar/dissimilar to CAPI?

Page 12 Study tool- Authors voice a concern in the introduction section, that digital platform COWIN (which was also facilitated by others and needed connectivity) could not have been used rural areas and then they use digital questionnaire on Survey Monkey. The enablers like surveyors, cache in the device, batch upload to a central server etc, need to be highlighted comparing the same with survey using printed questionnaire.

Reply: 

Sentence added, “The SurveyMonkey is a cloud-based online survey tool that can be emailed, sent through WhatsApp, posted on a website and shared on social media for self-administration.. It can be used for face-to-face data collection using a trained interviewer like CAPI. CAPI needs face-to-face interviews for data collection. The SurveyMonkey has a built-in basic data analysis package. Well-trained and experienced team who was involved in multiple community-based surveys at the national level. 

The SurveyMonkey© website is protected by TrustedSite software which actively monitors for security issues like malware, malicious links, and phishing. The website has a valid SSL certificate, and it uses SSL to encrypt communications with industry-standard security protocols

6. Page 12 Survey tool- was the tool itself translated in a vernacular language, was the consent part read in a vernacular language? How far was the interaction of the participants with the platform itself (particularly for consent), although it may not have been feasible in COVID-19 pandemic or post-pandemic.

Reply: 

Paragraph modified, 

The survey tool was developed in English language and was not translated into the local language. However, the patient information sheet (PIS), provided to each participant, was in the local language (HINDI). The study team explained and read aloud each question to eligible participants in their vernacular language (Hindi) during the interview. The survey team was well versed in English and was trained to ask questions in the local language.

Informed e-consent was obtained by asking the subjects about their willingness to participate, and the same was recorded on electronic forms. The electronic consent was used to avoid close contact between the participants and the survey team who visited door-to-door.”

7. Page 12 sample size and sampling technique- for an adult population above 18 years of age, 500 to 2000 count in 20 to 25 segments, each segment can have as low population as 25 or 50 and the range can be 25 to 100. Was there any contingency planned in case the sample size was not reached in 40 clusters, like enrolling more segments/clusters (at least planning done even if it was not needed eventually).

Reply:

Considering the average family size of Haryana (about 5), it is highly improbable that 70-100 adults will not be found in one cluster (having 20-25 households). 

The sentence added, “As a contingency plan, in case the adequate number of adults (70-100) was not present in one segment, adjoining the closest segment was also covered till the minimum number of 70 or more adults was reached.”

8. Page 12 Sample size calculation- revise the statement for clarity as- "One segment was selected randomly using the number of currency notes for inclusion in the survey" and the team covered all the households in the selected segment".

Reply:

Thank you to the reviewer for the suggestions.

It has replaced the previous sentence. 

9. Page 13 Study definitions- please clarify and add- "For the present study vaccine coverage was calculated for second dose".

Reply: 

Sentence removed.

10. Page 13 Ethics clearance- English correction- Please revise the statement as "Electronic consent was used to avoid close contact between the participants and the survey team who visited door-to-door".

Reply:

Done

11. Page 13 Data collection- How were the responses recorded in the survey monkey? What was done to ensure that there was no variation or difference in interpretation when different interviewers asked questions in their respective segments? What checks were deployed to ensure quality of data collection? Were there re-verification visits in a few households? Were there any measures taken within the Survey monkey platform?

Reply:

Sentence added, 

The data regarding infection with COVID-19 was collected based on self-reported positive testing for SARS-CoV 2 on reverse transcriptase polymerase chain reaction (RT-PCR). Diagnostic test reports of RTPCR were checked, if available; but the self-report was considered sufficient for the study. 

Since the infection was so unprecedented, we collected the infection history regardless of vaccination status after the demographic information. However, during the interview of the vaccination status, we categorically asked question regarding re-infection of COVID-19 after vaccination. The information on re-infection was only after vaccination, reinfection in the absence of vaccination was excluded. 

The survey tool was developed in English language and was not translated into the local language. However, the patient information sheet (PIS), provided to each participant, was in the local language (HINDI). The study team explained and read aloud each question to eligible participants in their vernacular language (Hindi) during the interview. The survey team was well versed in English and was trained to ask questions in the local language.

The data captured was directly integrated into a digital format, minimizing transcription errors that can occur with paper-based surveys. 

The survey team has experiences well in terms of community-based survey and have been involved in many national surveys using digital tool. The team was well versed in English language.

Further training of team members on the digital tool conducted before the survey and pretesting of the tool by the team was done to ensure the quality of data.”

Sentence added, “The supervisors were also responsible for re-verification visits in a few households, to ensure the quality of the data.”

12. Page 14 Data management and analysis- In case >20% of cells had an expected count of less than 5, what tests were used? Was there no plan for multivariate analysis like regerssion? Having a homogenous population from the same segment is not a valid justification for not planning regression, since the purpose here is to find out the association after cancelling the effect of each factor on another.

Reply; 

Sentence added, “To investigate the association between independent variables and dependent variables (vaccination status), chi-square statistics were used, and Fisher exact test was used when values in contingency table were <5. Multivariate regression analysis was used to find predictors of vaccine coverage, by using the socio-demographic factors, disability, and health status as independent variables.”

Table 4 with multivariate regression added in the results section.

13. Page 14 Results- Characteristics of the sample population- Please put a dot/full stop after (1619, Table 1).

Reply; 

Done

14. Page 14 Results- Characteristics of the sample population- In the line "Approximately, two quarters

of participants..." please replace the word culivators with retired.

Reply; 

Thanks to the reviewer for locating this typo error.

Sentence modified, “Majority of the respondents (59.2%, 1750) were currently not in the workforce, either unemployed, or retired or homemaker and 10.5% were cultivators by occupation, whereas few respondents worked in the public (3.4%, 100) or private (11.3%, 333) sectors. The retired, unemployed and homemaker participants were clubbed together as not working group. as in other COVID-19 vaccine study.”

15. Page 15 In the statement- "Considering the clinical characteristics..., please put % symbol after 4.9 and 3.2. This consistency needs to be checked throughout the manuscript

Reply: 

Thanks to the reviewer, 

Sentence modified, “Considering the pre-existing co-morbidities of the participants, hypertension (6.1%), joint problems (4.9%), and diabetes mellitus (3.2%) were the most common underlying self-reported health problems before suffering from COVID-19.”

16. Page 15 In the statement and throughout the manuscript- were the COVID-19 infections reported to have occured before vaccination or after. Similarly, for re-infections also, temporality needs to be explained- whether there was first infection before vaccination and another after or else-wise.

Reply:

Since the infection was so unprecedented, we collected the infection history regardless of vaccination status after demographic information. However, during the interview of the vaccination status, we categorically asked question regarding re-infection of COVID-19 after vaccination. 

The information on re-infection was only after vaccination, reinfection in the absence of vaccination was excluded. 

Definition added in methods section:

“Re-infection rate: Proportion of participants self-reporting RT-PCR positivity at least once within 12 months after receiving two doses of any COVID-19 vaccine (i.e. fully vaccinated).”

17. Page 15- Given the statements in the introduction about a higher prevalence and death in rural areas due to COVID-19, what can be the possible explanation regarding a low rate of COVID-19 infection found by the study as 6.2%?

Reply: 

We included persons reported a positive test for COVID-19 virus. Persons without a positive tested or not underwent for COVID-19 test were excluded. It is probable that individuals have COVID-19 symptoms who did not undergo testing, so further excluded them. 

Perhaps, this may be the reasons why the study found a low rate of COVID-19 disease. 

18. Page 17 COVID-19 vaccination status- for clarity, add the text- 100 participants received only one dose and 52 participants received no vaccine.

Reply: 

Sentence added, “In other words, 100 participants received only one dose, 2487 received two doses, 315 received three doses and 52 participants received no vaccine.”

19. Page 17 COVID-19 vaccination status- In the statement- Considering the COVID-19 virus disease..., please replace participants with fully vaccinated (grammatically wrong), with fully vaccinated participants.

Reply: 

Thanks to the reviewer. We have corrected it.

20. Page 18 In the statement- Persons with functional difficulties in vision- Revise the table number in the brackets as Table 2. Page 18 Table 3- Provide only significant p values in bold and other in non-bold font. Revise the column heading/cohort name of not vaccinated as - Not fully vaccinated- since unvaccinated are only 52, while 100 are partially vaccinated- hence the 100 partially vaccinated are not "not vaccinated".

Reply:

Thanks to the reviewer. We have corrected it.

21. In first line of Discussion- Page 19- remove the left over "t".

Reply: done

22. To put the things in the perspective, add to the discussion- the reported national coverage for rural population?

Reply:

The national coverage is not available separately for urban and rural areas on the CoWIN dashboard.

23. Page 19- Discussion section- In the statement- Hence, the present study shows that the coverage... revise the text as- at the same time, the coverage for the second dose is better than the state average.

Reply:

Thanks to the reviewer. We have corrected it.

24. Page 19, the statement "The proportion of vaccines being received found..." the statement mentions corresponding values but the same is missing for Pfizer vaccine, and can we name this vaccine the way we have done for other vaccines?

Reply;

Sentence modified, “The proportion of vaccines administered free of cost by the government are Covishield (79.3%), Covaxin (16.5%), Sputnik V (0.06%), Corbevax (3.3%) and Covovax (0.002%). Pfizer was available only in the private sector in India and in selected states only due to its higher cost and regulatory restrictions.”

25. Discussion- global comment- clarity is needed regarding the post-vaccination infections and second episodes of infection. Also need to explain the value addition of the study besides finding out vaccine coverage. Also, was it highlighted that the seroprevalence studies undertaken by other agencies will be affected by vaccine coverage as well. Temporality of the infections and vaccinations need to be clarified even if these participants were fully vaccinated. Another thing is highlighting the comparison between unvaccinated, partially vaccinated, and fully vaccinated individuals.

Reply:

Thanks to the reviewer for the suggestions. We have improved the discussion in the relevant part.

26. Page 20- comparison of the study with other studies- How do we compare the results of this study with the following systematic review with pooled seroprevalence of 20 to 70 %?

Reply:

We agreed with the reviewer. For seroprevalence study, even vaccinated persons will be taken into consideration. Further, the statement is not about the comparison. Since there are no similar studies available in India it is worth highlighting a few pieces of evidence on the prevalence of COVID-19 irrespective of case definition. We have revised it.

27. Jahan N, Brahma A, Kumar MS, Bagepally BS, Ponnaiah M, Bhatnagar T, Murhekar MV. Seroprevalence of IgG antibodies against SARS-CoV-2 in India, March 2020 to August 2021: a systematic review and meta-analysis. Int J Infect Dis. 2022 Mar;116:59-67. doi: 10.1016/j.ijid.2021.12.353. Epub 2021 Dec 28. Erratum in: Int J Infect Dis. 2022 Jun;119:119. PMID: 34968773; PMCID: PMC8712428.

Reply: 

Reference removed

28. Page 20- Put % sign in the brackets in the statement- joint problems (4.9) and diabetes mellitus (3.2) being the most common self-reported comorbidities.

Reply:

Thanks to the reviewer. We have corrected it.

29. Page 21- In the statement- To mitigate this inequity in coverage...- What were the barriers found out in the current survey for vaccine coverage in PwDs? Was there any qualitative data collected in the survey and can we at least highlight the barriers found?

Reply:

Very nice inputs and important.

The information on the barriers to vaccine coverage in PWDs was not a part of our study. However, we plan further study such as qualitative or semi-structured interviews in the future to rule out the barriers.

Sentence modified, “. To mitigate this inequity in coverage, further studies are needed such as qualitative study or semi-structured interviews to identify the barriers that PwDs face in accessing vaccination and determine the appropriate strategies to address poor coverage.”

30. Page 21- a statement of not finding the significance for any factor needs to be made regarding association. Please refer to the earlier comments.

Reply:

Multivariate analysis and associated paragraphs have been added in results and discussion.

“Gender was not a significant predictor of vaccine coverage in the current study; previous studies had reported differences in coverage based on gender, but concluding evidence to determine gender effect is lacking. Older age and higher education were found to be significant predictors in this rural study area, which is also corroborated by previous studies”

31. Page 22- Conclusions- in the first statement- remove "of" from the statement- viewed as the most important public health measure of against the SARS-CoV 2.

Reply: 

Done 

Reviewer 5.

1. As per the topic name of the article: Vaccination Coverage against COVID-19 among a Rural Population: A Cross, Sectional Study in a Northern Part of India, the more focus should have been given to the village conditions like transportation, electricity, restrictions etc. that differentiate the village and urban infrastructure and highly affect accessibility to basic health care services and the vaccination coverage. Also cross check the title of the article as you are saying it is the cross sectional study but your retracting the data of the past in this study that is objective of case control study.

Reply: 

Thanks to the reviewer. 

Differentiating the factors that can affect the vaccination drive between rural and urban areas is an important aspect to study. At present, our focus is only rural areas. In the future, we plan to cover both areas and thereby compare it and factors affecting the achievement. 

The study is cross-sectional because the coverage, infection rate and re-infection rate were determined at one point in time only. The study made no comments on the causal association between vaccination and infection. 

Limitation added, “In addition, temporal associations between vaccination and infections cannot be deduced owing to the cross-sectional nature of the study, which might require a case-control study design.”

2. This study is conducted in a very small geographic area so the significance of the results cannot be generalized.

Reply:

Thank you to the reviewer. This is one of the limitations and has been added in the revised manuscript as highlight in the manuscript. 

3. The base seroprevalance for sample size calculation in the study was 8.5% , there is a need to specify the source or reference of this seroprevalence of COVID-19 in the adult population, as mentioned in the subheading "Sample size and Sampling technique". In the discussion part, you have mentioned the sero prevalence being 0.73 % in India. There seems a mismatch in both the statements and needs clarification.

Reply

The sample size is calculated based on the seroprevalence in rural areas (reference 16).

Discussion added, “The national COVID-19 serosurvey was a large community-based study conducted by the Indian Council of Medical Research (ICMR) among the general population, which reported population-weighted seroprevalence of 0.73% [95% CI: 0.34-1.13] in May 2020, and further increased to 67.6% (95% CI: 66.4- 68.7) by July 2021.[30,31] This indicates seroconversion owing to the effect of unprecedented spread of natural infection as well as vaccination.”

4. In Covid 19 Vaccine status subheading, it is mentioned that "Covishield was the type of COVID-19 vaccine that had been received by most participants (81.3%, 2380) followed by Covaxin (12.3%, 361) and Pfizer (0.03, 1)" . The source of Pfizer Vaccine should be mentioned as it was not granted clearance by Indian regulatory authorities.

Reply:

Sentence added, “Pfizer was available only in the private sector in India and in selected states only due to its higher cost and regulatory restrictions.”

5. A comparison between the vaccination coverage (full and partial) at the national and state (Haryana) level data available on CoWin dashboard on 28th June 2023 is made with the data of the study. It is better to also compare the study data with the CoWin Dashboard vaccination coverage of study area. The reasons for the difference found, if any might be speculated.

Reply

Thank you to the reviewer. 

The Cowin dashboard provides data up to state level only, not district level. However, we have mentioned data of the state as a part of the discussion though there is no available data for district level.

6. The vaccination coverage in study area is better than state as well as national level coverage. So, various steps taken up by authorities like mass campaigns, setups, infrastructure and protocols for COVID19 vaccination in your study area could be compared and explained in your article. this would be act as a reference for other disease vaccination programmes and vaccination programmes to be conducted in other areas too.

Reply

Thank you to the reviewer for the suggestions. Further study is required to investigate why this study area has better level of coverage than the state concerned and national level. Findings from such studies may help to address other vaccination program. 

However, several factors may account for the better level of coverage, for example, good healthcare infrastructure in Jhajjar, Haryana, including a national dedicated COVID-19 care facilities along free vaccination services, active mass campaign, good connectivity leading to better transportation perhaps improve the coverage. 

Relevant texts have been added. 

7. The statement on the last paragraph of page no 13, the vaccine status of current study was compared with national status and corresponding values were mentioned viz “The proportion of vaccines being received found in the current study was Covaxin (12.3%), Covishield (81.3%) and Pfizer (0.03%). The corresponding values for the national level are Covishield (79.3%), Covaxin (16.5%) and Sputnik V (0.06%) respectively. Corresponding values for sputnik v in the study as well as Pfizer vaccine at national level should be mentioned.

Reply

Sentence modified, “The proportion of vaccines administered free of cost by the government are Covishield (79.3%), Covaxin (16.5%), Sputnik V (0.06%), Corbevax (3.3%) and Covovax (0.002%). Pfizer was available only in the private sector in India and in selected states only due to its higher cost and regulatory restrictions.”

8. A clarification regarding why in the study, there is reporting of the reinfection of COVID 19 among only fully vaccinated people, that too only after 12 months or more, post vaccination. The data regarding reinfection before 12 months post vaccination should also be mentioned. The comparative statement of difference between reinfection after partial vaccination and full vaccination would also be useful and help in estimating the relative Risk between these two groups.

Reply

Sentence added, “The duration of 12 months between the second vaccine dose and re-infection was stipulated to avoid any overlap between post-COVID-19 syndrome (especially long COVID) and re-infection. Previous studies have demonstrated that long COVID usually starts after 3 months of infection, and the sequelae can persist even after 6 months.”

9. Reinfection rate as per type of vaccine should also be mentioned.

Reply

The number of participants reporting one or multiple reinfections was 46 & 5 respectively. These numbers are too small to be further segregated by vaccine types. So, we present re-infection rate post vaccination regardless types of vaccine.

However, further study with a large sample size with help to investigate re-infection rate according to the types of vaccine.

10. . The vaccine effectiveness in terms of reduced COVID19 cases and deaths pre and post vaccination should be taken up in the study.

Reply

We regret that the current study was not aimed at measuring vaccine effectiveness, for which plethora of evidence is already in the literature. Similarly, measurement of mortality data was also not included in the study.

11. There is a difference of 3.4 % between partial vaccination and full vaccination, the data regarding various reasons for partial vaccination should be mentioned (like: as mentioned in the article that the hesitancy for covid 19 vaccination is more in urban population compared to rural population).

Reply 

Done

12. The proportion of partially vaccinated is 3.4 %, the data regarding theses partially vaccinated people should be correlated with various variables like sex, occupation, educational status and disease literacy, etc

Reply

Table 4 on multiple regression is added in which the 100 persons partially vaccinated and 52 persons not vaccinated have been clubbed together to identify predictors of vaccine coverage.

13. The reinfection rate of 1.6% for single infection and 0.2 % of multiple infection is reported in the study, is there any underlying factor (health problems or occupation) for the reinfections reported.

Reply

The number of participants reporting one or multiple reinfections was 46 & 5 respectively. These numbers are too small to be further segregated by health problems or occupation. However, further study is warranted to investigate factors leading to re-infection following vaccination in a larger sample.

14. One of the major limitation is that it solely dependent on the individual response that is susceptible for recall biasness and social desirability bias, so you have to give an explanation of the method you adopted for removal or reduction of these biasness in your data.

Reply

Limitation added, “Third, social desirability and recall biases are inherent due to the self-reported nature of the study.”

15. The data regarding aefi, if any should also be included in the study.

Reply 

We regrated that AEFI data was not collected as part of the study. But few participants reported skin discoloration, fever, feeling of general weakness after vaccination. 

But we did not document this information. 

18. The time line in which the vaccination has been achieved in rural areas and urban areas should be compared.

Reply 

The study was conducted from January to February 2023, much after the rollout of universal COVID-19 vaccination, including in Haryana in 2021.

19. Please correct Reference no 5., 19, 20, 22, 23, 24; as per journal guidelines.

Reply 

Done

---

## [Decision Letter · Decision Letter 1]

9 Jan 2024

PONE-D-23-23474R1Vaccination Coverage against COVID-19 among Rural Population in Haryana, India: A Cross-Sectional StudyPLOS ONE

Dear Dr. Senjam,

Thank you for submitting your manuscript to PLOS ONE. After careful consideration, we feel that it has merit but does not fully meet PLOS ONE’s publication criteria as it currently stands. Therefore, we invite you to submit a revised version of the manuscript that addresses the points raised during the review process.

We look forward to receiving your revised manuscript.

Kind regards,

Paavani Atluri

Academic Editor

PLOS ONE

Journal Requirements:

Reviewers' comments:

Reviewer's Responses to Questions

**Comments to the Author**

1. If the authors have adequately addressed your comments raised in a previous round of review and you feel that this manuscript is now acceptable for publication, you may indicate that here to bypass the “Comments to the Author” section, enter your conflict of interest statement in the “Confidential to Editor” section, and submit your "Accept" recommendation.

Reviewer #1: All comments have been addressed

Reviewer #5: All comments have been addressed

Reviewer #10: (No Response)

Reviewer #12: All comments have been addressed

Reviewer #13: All comments have been addressed

2. Is the manuscript technically sound, and do the data support the conclusions?

Reviewer #1: Yes

Reviewer #5: Yes

Reviewer #10: Partly

Reviewer #12: Yes

Reviewer #13: Yes

3. Has the statistical analysis been performed appropriately and rigorously? 

Reviewer #1: Yes

Reviewer #5: Yes

Reviewer #10: (No Response)

Reviewer #12: Yes

Reviewer #13: Yes

4. Have the authors made all data underlying the findings in their manuscript fully available?

Reviewer #1: Yes

Reviewer #5: Yes

Reviewer #10: (No Response)

Reviewer #12: Yes

Reviewer #13: Yes

5. Is the manuscript presented in an intelligible fashion and written in standard English?

Reviewer #1: Yes

Reviewer #5: Yes

Reviewer #10: (No Response)

Reviewer #12: Yes

Reviewer #13: Yes

6. Review Comments to the Author

Reviewer #1: (No Response)

Reviewer #5: Dear Authors, Good that many suggested changes are incorporated and revised.

small changes:

page 6 study tool section: More importance is given to data management platform- survey monkey but study tool is the questionnaire used in study. This will create ambiguity for the readers.

Page 7- Pilot testing is mentioned but number or percentage of pilot tested population is missing.

Reviewer #10: (No Response)

Reviewer #12: (No Response)

Reviewer #13: Dear Author,

All the comments and suggestion are well addressed and the manuscript is written well. However you should also discuss more about the factors like set up of vaccination program, factors, coordination, people awareness etc which may explain this large vaccination coverage for Covid19.

7. PLOS authors have the option to publish the peer review history of their article (what does this mean?). If published, this will include your full peer review and any attached files.

Reviewer #1: No

Reviewer #5: No

Reviewer #10: No

Reviewer #12: No

Reviewer #13: **Yes: **Baleshwari Dixit

---

## [Author Response · Author response to Decision Letter 1]

17 Jan 2024

Responses to the comments 

The authors would like to thank the editor and all reviewers for their time spent reading the manuscript and for helping us to improve the manuscript further. We have revised it as per advice. The revisions are highlighted.

Comments 

Reviewer #5: Dear Authors, Good that many suggested changes are incorporated and revised.

page 6 Study tool section: More importance is given to the data management platform- Survey Monkey but the study tool is the questionnaire used in the study. This will create ambiguity for the readers.

Page 7- Pilot testing is mentioned but the number or percentage of pilot tested population is missing.

Reply

Thank you, reviewer. You are right. SurveyMonkey is not a survey tool, rather it is a software (cloud-based) that can be used to develop survey questionnaires. Correction has been done on page 7. During the pilot testing, we covered 10 Households ( 31 eligible participants) 

Comments 

Reviewer #13: Dear Author,

All the comments and suggestion are well addressed and the manuscript is written well. However you should also discuss more about the factors like set up of vaccination program, factors, coordination, people awareness etc which may explain this large vaccination coverage for Covid19.

Reply

We are grateful to you for your advice and suggestions. We have incorporated the suggested points on page 26. We also added about CoWIN app since it is easy to use and user friendly.

---

## [Decision Letter · Decision Letter 2]

13 Feb 2024

Vaccination Coverage against COVID-19 among Rural Population in Haryana, India: A Cross-Sectional Study

PONE-D-23-23474R2

Dear Dr. Suraj Singh Senjam,

We’re pleased to inform you that your manuscript has been judged scientifically suitable for publication and will be formally accepted for publication once it meets all outstanding technical requirements.

Kind regards,

Paavani Atluri

Academic Editor

PLOS ONE

Additional Editor Comments (optional):

Reviewers' comments:

Reviewer's Responses to Questions

**Comments to the Author**

1. If the authors have adequately addressed your comments raised in a previous round of review and you feel that this manuscript is now acceptable for publication, you may indicate that here to bypass the “Comments to the Author” section, enter your conflict of interest statement in the “Confidential to Editor” section, and submit your "Accept" recommendation.

Reviewer #10: All comments have been addressed

Reviewer #12: All comments have been addressed

2. Is the manuscript technically sound, and do the data support the conclusions?

Reviewer #10: Yes

Reviewer #12: Yes

3. Has the statistical analysis been performed appropriately and rigorously? 

Reviewer #10: I Don't Know

Reviewer #12: Yes

4. Have the authors made all data underlying the findings in their manuscript fully available?

Reviewer #10: Yes

Reviewer #12: Yes

5. Is the manuscript presented in an intelligible fashion and written in standard English?

Reviewer #10: Yes

Reviewer #12: Yes

6. Review Comments to the Author

Reviewer #10: (No Response)

Reviewer #12: Well written article and Information about Covid vaccination is very crucial in the present situation.

7. PLOS authors have the option to publish the peer review history of their article (what does this mean?). If published, this will include your full peer review and any attached files.

Reviewer #10: No

Reviewer #12: No

---

## [Editor Report · Acceptance letter]

27 Feb 2024

PONE-D-23-23474R2 

PLOS ONE

Dear Dr. Senjam, 

I'm pleased to inform you that your manuscript has been deemed suitable for publication in PLOS ONE. Congratulations! Your manuscript is now being handed over to our production team.

Kind regards, 

on behalf of

Dr. Paavani Atluri 

Academic Editor

PLOS ONE